# Information-Guided Diffusion Sampling for Image Classification Dataset Distillation

## Abstract

Image classification dataset distillation aims to create a compact dataset that retains essential information while maintaining model performance. Diffusion models (DMs) have shown promise for this task but struggle in low images-per-class (IPC) settings, where generated samples lack diversity. In this paper, we address this issue from an information-theoretic perspective by identifying two key types of information that a distilled dataset must preserve: (*i*) *prototype information* $\mathrm{I}(X;Y)$, which captures label-relevant features; and (*ii*) *contextual information* $\mathrm{H}(X|Y)$, which preserves intra-class variability. Here, $(X, Y)$ represents the pair of random variables corresponding to the input data and its ground truth label, respectively. Observing that the required contextual information scales with IPC, we propose maximizing $\mathrm{I}(X;Y) + \beta \mathrm{H}(X|Y)$ during the DM sampling process, where $\beta$ is IPC-dependent. Since directly computing $\mathrm{I}(X;Y)$ and $\mathrm{H}(X|Y)$ is intractable, we develop *variational estimations* to tightly lower-bound these quantities via a data-driven approach. Our approach, information-guided diffusion sampling (IGDS), seamlessly integrates with diffusion models and improves dataset distillation across all IPC settings. Experiments on Tiny ImageNet and ImageNet subsets show that IGDS significantly outperforms existing methods, particularly in low-IPC regimes. The code is available at `https://anonymous.4open.science/r/IGDS-4C0F/`.

## 1 Introduction

The success of high-performance deep neural networks (DNNs) is largely attributed to large-scale, highly informative datasets (LeCun et al., 2015). However, the size of these datasets poses a substantial burden on storage and computational resources during model training (Deng et al., 2009; Salamah et al., 2024; Kaplan et al., 2020). To mitigate the cost of training DNNs, dataset distillation (Wang et al., 2018; Sachdeva & McAuley, 2023) has been extensively studied in recent years as a potential solution to compress datasets, thereby reducing both storage requirements and computational costs. In this approach, a smaller dataset whose compactness is typically measured by images-per-class (IPC) is constructed as a substitute for the original dataset, while still enabling the trained model to achieve decent generalization performance on unseen test data points.

To construct such a compact dataset, the distillation process typically involves an iterative optimization of pixel values and auxiliary model weights to align either with the model's weight trajectory (Zhao et al., 2021; Cazenavette et al., 2022b) or feature statistics (Wang et al., 2022; Deng et al., 2024; Sajedi et al., 2023). However, this approach has two major drawbacks: (*i*) High computational cost—most existing methods require jointly optimizing auxiliary model parameters and distilled samples at the pixel level through an iterative process, resulting in significant computational overhead. (*ii*) Poor generalization across different model architectures—the performance of models trained on the distilled dataset is highly dependent on the architecture of the auxiliary DNN. When the target model's architecture differs from that of the auxiliary DNN, significant performance degradation is often observed.

To overcome these drawbacks, recent studies have proposed generative distillation (Zhao & Bilen, 2022; Cazenavette et al., 2023), which leverages a generative model to synthesize a new, compact dataset. In this approach, a generative model is trained on the target dataset and then used to sample distilled data (Zhao &

Bilen, 2022; Li et al., 2024). Consequently, the resulting distilled dataset is both model-architecture-agnostic and more efficiently generated. Among generative models, diffusion models (DMs) have emerged as a strong choice for dataset distillation, demonstrating state-of-the-art performance in this setting (Gu et al., 2024b; Su et al., 2024b; Chen et al., 2025; Li et al., 2025). Nonetheless, DM-based dataset distillation suffers from poor performance in low-IPC scenarios, where the number of IPCs is small. In these cases, the accuracy is nearly as low as training on a randomly chosen subset. A primary reason is that, under low-IPC conditions, the model's generated samples reflect only part of the true data distribution, leading to a distilled dataset with limited diversity and substantial information loss. This shortfall grows more severe as the IPC decreases.

To address this limitation and enable the generation of informative samples, we first seek to quantify the essential information that must be preserved. To this end, we adopt an information-theoretic perspective (Shannon, 1948; Cover & Thomas, 2006; Yang et al., 2024). Specifically, we quantify the relevant information using the Shannon entropy $\mathrm{H}(\cdot)$ (Shannon, 1948) on the random variable (RV) $X$, which represents the input dataset. We then expand $\mathrm{H}(X)$ as: $\mathrm{H}(X) = \mathrm{I}(X;Y) + \mathrm{H}(X|Y)$, where $Y$ is an RV denoting the ground-truth (GT) label [1]. Through this decomposition, the total information in $\mathrm{H}(X)$ naturally splits into: (*i*) *prototype information* $\mathrm{I}(X;Y)$, reflecting how much information $X$ provides about its GT label; and (*ii*) *contextual information* $\mathrm{H}(X|Y)$, capturing the remaining information in $X$ once its GT label is given.

A successful dataset distillation should preserve both the prototype and contextual information of the target dataset. However, we observe that the required amount of contextual information depends on the IPC: a higher (resp. lower) IPC necessitates more (resp. less) contextual information. Building on this insight, we propose to maximize $\mathrm{I}(X;Y) + \beta\mathrm{H}(X|Y)$ during the DM sampling process, where $\beta$ is selected according to the IPC. Since directly computing $\mathrm{I}(X;Y)$ and $\mathrm{H}(X|Y)$ is intractable, we develop *variational estimations* to tightly lower-bound these quantities via a data-driven approach. Specifically, we train a DNN using a novel training algorithm that provides tight lower bounds on both $\mathrm{I}(X;Y)$ and $\mathrm{H}(X|Y)$. We refer to this DNN as a variational estimator (VE). Once the VE is trained, it is frozen and used in the DM sampling process, guiding the generation of distilled data that maximally preserves both prototype and contextual information.

Our work introduces a novel dataset distillation approach, Information-Guided Diffusion Sampling (IGDS), which leverages information-theoretic principles to enhance the effectiveness of diffusion models in low IPC settings. The key contributions of this paper are as follows:

● We introduce a principled framework based on Shannon entropy decomposition, identifying prototype information $\mathrm{I}(X;Y)$ and contextual information $\mathrm{H}(X|Y)$ as crucial components for effective dataset distillation. Our approach dynamically balances these terms to optimize the informativeness of the distilled dataset.

● Since directly computing prototype information $\mathrm{I}(X;Y)$ and contextual information $\mathrm{H}(X|Y)$ is intractable, we develop a data-driven VE using deep neural networks to obtain tight lower bounds on these quantities. This estimator is seamlessly integrated into the diffusion sampling process.

● We observe that the optimal amount of contextual information varies across different IPC settings. Motivated by this observation, we propose IGDS, a novel diffusion-based dataset distillation method that maximizes $\mathrm{I}(X;Y) + \beta\mathrm{H}(X|Y)$ during the sampling process. The weight $\beta$ is IPC-dependent, allowing for adaptive control over prototype and contextual information retention.

● Extensive experiments on Tiny ImageNet (Le & Yang, 2015) and subsets of ImageNet (Deng et al., 2009) demonstrate that IGDS achieves superior performance compared to existing methods, particularly in low-IPC scenarios, where prior diffusion-based approaches suffer from poor diversity and high information loss.

## 2 Related Works

Dataset distillation has received widespread attention since it was proposed, and a substantial amount of research has contributed to its rapid development (Li et al., 2022b; Yu et al., 2023). The proposed methods

---

[1]Hereafter, For brevity, we use prototype information to denote the mutual information between the input and the ground-truth label, $\mathrm{I}(X;Y)$, and contextual information to contextual information to denote the conditional entropy of the input given its ground-truth label $\mathrm{H}(X \mid Y)$.

can be classified into non-generative and generative approaches. We defer discussion of related information-theoretic work on neural networks to Appendix B.

### 2.1 Non-generative Dataset Distillation

This approach initializes the distilled dataset and optimizes the images during the distillation process using specific algorithms such as matching-based and kernel-based methods (Nguyen et al., 2021a). Matching-based methods ensure the ability to distill by matching parameters, features, or distributions between the original and distilled datasets. For instance, methods like DC (Zhao & Bilen, 2021) and IDC (Kim et al., 2022a) match the gradient obtained by training both on original and synthetic data. While approaches like MTT (Cazenavette et al., 2022a) and ATT (Liu et al., 2024) achieve parameter matching by minimizing the loss over the training trajectory on original and synthetic data. Kernel-based methods take ridge regression as the optimization objective and employ a neural tangent kernel to generate distilled datasets (Nguyen et al., 2021b).

### 2.2 Generative Dataset Distillation

This method distills the knowledge of the original dataset into the generative model, which is then used to obtain distilled datasets in the subsequent sampling phase (Zhao & Bilen, 2022; Cazenavette et al., 2023). In each distillation process, traditional methods optimize one dataset of a specific size, as defined by the IPC. Generative methods, however, can generate any number of datasets of any size (Gu et al., 2024a; Su et al., 2024a;b). This flexibility makes dataset distillation free from the constraints of IPC, thus saving a significant amount of time when distillation needs to be executed more than once, which is common in various downstream tasks of dataset distillation, such as continue learning (Yang et al., 2023a), privacy preservation (Li et al., 2020; 2022a), and neural architecture search (Ding et al., 2024).

Among generative approaches, both our proposed IGDS and the recent influence-guided diffusion (IGD) method (Chen et al., 2025) utilize diffusion models, but differ substantially in motivation and methodology. IGD is training-free and guides sampling based on influence functions that quantify downstream model impact. In contrast, IGDS adopts an information-theoretic perspective, maximizing mutual and conditional entropy through variational estimators. This enables IGDS to better preserve diversity under low-IPC constraints, whereas IGD is more effective in high-IPC regimes.

### 2.3 Diffusion Model

Having a certain degree of creativity, generative models have experienced rapid advancement in recent years. Generative approaches like GANs and VAEs have achieved broad application in various industries. Among them, diffusion models like Imagen (Saharia et al., 2023) and Stable Diffusion (Rombach et al., 2022) have gained significant attention. They acquire the ability to recover an image from random noise by learning how to predict noise from noisy images, which are widely recognized for their stability and effectiveness across distillation tasks. Specific to the field of computer vision, they have been proven effective in a wide range of scenarios (Yang et al., 2023b). For instance, LDM (Rombach et al., 2022) enables the diffusion process in the latent space to save computational resources. DDeP (Brempong et al., 2022) utilizes text-to-image diffusion models to obtain promising results on semantic segmentation. Palette (Saharia et al., 2022) tackles several image generation tasks with conditional diffusion models, and FDM (Harvey et al., 2022) allows for sampling specific video frames from other video subsets. Our method aims to explore the potential of the diffusion model in dataset distillation.

## 3 Notation

For a positive integer $C$, let $[C] \triangleq \{1, \ldots, C\}$. Denote by $P[i]$ the $i$-th element of the vector $P$. For two vectors $U$ and $V$, denote by $\langle U, V \rangle$ their inner product. For two matrices $M \in \mathbb{R}^{m \times n}$ and $N \in \mathbb{R}^{n \times k}$, denote by $M \cdot N$ their matrix product. We use $|\mathcal{C}|$ to denote the cardinality of a set $\mathcal{C}$. The entropy of $C$-dimensional

$$Y \xrightarrow[\substack{\text{Sampling according} \\ \text{to } P_{X|Y}(\cdot\,|y)}]{} X \xrightarrow[\text{Encoder } f_{\boldsymbol{\theta}}(\cdot)]{} \hat{X} \xrightarrow[\text{Classifier } g_{\boldsymbol{\psi}}(\cdot)]{} \hat{Y}$$

Figure 1: Multi-class classification can be modeled as a Markov chain. Sample $X$ is sampled from the class $Y$, according to the $P_{X|Y}(\cdot|y)$. The encoder then maps the $X$ to the feature representation $\hat{X}$, which is further mapped by the classifier to an output probability vector $\hat{Y}$.

probability vector $P$ is defined as $\mathrm{H}(P) = \sum_{c=1}^{C} -P[c] \log P[c]$. Also, the Kullback–Leibler (KL) divergence of two $C$-dimensional probability vectors $P_1$ and $P_2$ is defined as $\mathrm{KL}(P_1||P_2) := \sum_{c=1}^{C} P_1[c] \log \frac{P_1[c]}{P_2[c]}$.

For a random variable $X$, denote by $P_X$ its probability distribution, and by $E_X[\cdot]$ the expected value w.r.t. $X$. For two random variables $X$ and $Y$, denote by $\mathbb{P}_{(X,Y)}$ their joint distribution. The mutual information between two random variables $X$ and $Y$ is defined as $\mathrm{I}(X;Y) = \mathrm{H}(X) - \mathrm{H}(X|Y)$, and the conditional mutual information of $X$ and $Y$ given a third random variable $Z$ is $\mathrm{I}(X;Y|Z)$. The softmax operation is denoted by $\sigma(\cdot)$.

Consider an image classification dataset $\mathcal{D}$ of size $n$ with $C$ classes, $\mathcal{D} = \{(\boldsymbol{x}_i, y_i)\}_{i=1}^{n}$, sampled from an unknown distribution $P_{(X,Y)}$, where each $\boldsymbol{x}_i \in \mathbb{R}^d$ and $y_i \in [C]$. For any class $y$, we define $\mathcal{D}_y = \{(\boldsymbol{x}_j, y_j) \in \mathcal{D}|y_j = y\}$ the subset of $\mathcal{D}$ containing all samples with label $y$.

For a prescribed images-per-class (IPC) budget $k$, the goal of dataset distillation is to construct a smaller dataset $\mathcal{S}$ such that training a DNN $f$ from scratch on $\mathcal{S}$ yields a model whose performance closely matches that of a model trained on the original dataset $\mathcal{D}$, ideally across a target family of architectures.

## 4 Methodology

As discussed in section 1, the performance of DM-based dataset distillation degrades significantly when the IPC is small. In such low-IPC conditions, the DM tends to produce samples that represent only a portion of the true data distribution, omitting many essential modes. As a result, the distilled dataset exhibits limited diversity and loses substantial information about the underlying classes. Empirically, this often manifests in accuracies that are comparable to training on a mere random subset of the original dataset. The challenge intensifies with decreasing IPC, since fewer examples per class mean the generative process has even less guidance for reproducing the full range of relevant features. In practice, these shortcomings severely limit the applicability of DM-based distillation for real-world tasks where one cannot afford to collect a large number of samples for each class.

To tackle this limitation, it is crucial to identify the core information that must be retained in the distilled dataset $\mathcal{S}$. To this end, we adopt an information-theoretic perspective to rigorously quantify and preserve this information. Specifically, we measure the total information in the input dataset, represented as a random variable $X$, using the Shannon entropy $\mathrm{H}(X)$. Noting that a significant part of the value of a dataset lies in its ability to determine labels, we decompose $\mathrm{H}(X)$ as follows

$$\mathrm{H}(X) = \underbrace{\mathrm{I}(X;Y)}_{\text{prototype information}} + \underbrace{\mathrm{H}(X|Y)}_{\text{contextual information}}, \tag{1}$$

where $Y$ is a random variable denoting the GT label. This decomposition distinctly separates prototype information $\mathrm{I}(X;Y)$, which quantifies how much $X$ reveals about its label, from contextual information $\mathrm{H}(X|Y)$, which captures the variability and richness of the data given its label. In other words, prototype information ensures that the distilled dataset remains discriminative for classification tasks, whereas contextual information guards against the collapse into a narrow subset of features, thus maintaining diversity and nuance (in section 5.3, we visualize the semantic meaning of prototype and contextual information). By explicitly accounting for both these components, we can better capture the data's essential characteristics, even in low-IPC regimes. A successful dataset distillation scheme must retain both prototype information,

ensuring that each class is accurately characterized, and contextual information, preserving the variety and richness of the underlying data distribution. However, our observations indicate that the requisite amount of contextual information scales with the IPC: higher IPC scenarios allow, and indeed necessitate, more contextual detail, whereas lower IPC settings benefit more from a tighter focus on prototype information (please see section 5.2 for additional details).

Guided by this insight, we aim to balance these two information types by maximizing the objective

$$\mathrm{I}(X;Y) \,+\, \beta\,\mathrm{H}(X|Y), \qquad (2)$$

where the scalar $\beta > 0$ is chosen to reflect the IPC: a larger $\beta$ for high-IPC settings increases the emphasis on contextual richness, while a smaller $\beta$ in low-IPC scenarios prioritizes critical prototype information.

Computing $\mathrm{I}(X;Y)$ and $\mathrm{H}(X|Y)$ is challenging, and to the best of our knowledge, no previous work has accomplished this. To overcome this difficulty, we introduce a novel method in section 4.1 that provides variational estimates for these quantities. Subsequently, in section 4.2, we leverage these estimates to guide the sampling process of diffusion models.

---

**Algorithm 1** Pseudo-code for Training the $f_{\boldsymbol{\theta}}(\cdot)$

1: **Input:** $f_{\boldsymbol{\theta}}, f_{\boldsymbol{m}}$: initialized encoder and momentum encoder, *queue*: dictionary as a queue of $K$ keys, $m$: momentum, *aug*: random augmentation method, $\tau$: temperature and $\lambda > 0$.
2: $f_{\boldsymbol{\theta}}$.params $= f_{\boldsymbol{m}}$.params
3: **for** $x \in D$ **do**
4: $\quad x_q, x_k = \mathrm{aug}(x), \mathrm{aug}(x)$
5: $\quad q, k = f_{\boldsymbol{\theta}}(x_q), f_{\boldsymbol{m}}(x_k)$.detach()
6: $\quad H_q, H_k = \mathrm{softmax}(q), \mathrm{softmax}(k)$
7: $\quad Q = (H_q + H_k)/2$
8: $\quad l_{pos}, l_{neg} = \langle q, k \rangle, q \cdot k^T$
9: $\quad \mathrm{logits} = \mathrm{cat}([l_{pos}, l_{neg}], \mathrm{dim}{=}1)$
10: $\quad \mathrm{labels} = \mathrm{zeros}(\mathrm{N})$
11: $\quad \mathrm{loss} = \mathrm{CE}\ (\mathrm{logits}\ /\ \tau, \mathrm{labels})\ \text{-}\ \lambda\mathrm{KL}(H_q||Q^Y)$
12: $\quad \mathrm{loss.backward}()$
13: $\quad \mathrm{update}(f_{\boldsymbol{\theta}}.\mathrm{params})$
14: $\quad f_{\boldsymbol{m}}$.params $= \mathrm{m} \times f_{\boldsymbol{m}}$.params $+ (1\text{-m}) \times f_{\boldsymbol{\theta}}$.params
15: $\quad \mathrm{enqueue}(\mathrm{queue}, k)$
16: $\quad \mathrm{dequeue}(\mathrm{queue})$
17: **end for**
18: **Output:** $f_{\boldsymbol{\theta}}$

---

*The proofs for all propositions presented in this paper are deferred to section C.*

### 4.1 Variational Estimates for $\mathrm{I}(X;Y)$ and $\mathrm{H}(X|Y)$

We employ an auxiliary DNN composed of an encoder $f_{\boldsymbol{\theta}}(\cdot)$ and a classifier $g_{\boldsymbol{\psi}}(\cdot)$ to help us finding variational estimates for both $\mathrm{I}(X;Y)$ and $\mathrm{H}(X|Y)$. The encoder transforms the input $X$ into a feature representation $\hat{X}$, and the classifier maps $\hat{X}$ to a probability vector $\hat{Y}$. In this setup, the random variables $\{Y, X, \hat{X}, \hat{Y}\}$ form a Markov chain in the order shown in fig. 1 (see Yang et al. (2025) for more details). Now, in section 4.1.1 and section 4.1.2, we show how this auxiliary DNN can be leveraged to derive variational estimates for $\mathrm{I}(X;Y)$ and $\mathrm{H}(X|Y)$, respectively. Then, in section 4.1.3, we train $f_{\boldsymbol{\theta}}(\cdot)$ to give us the estimates.

#### 4.1.1 Variational Estimates for $\mathrm{I}(X;Y)$

Equipped with $f_{\boldsymbol{\theta}}(\cdot)$ and classifier $g_{\boldsymbol{\psi}}(\cdot)$, in this section, we propose a method to find a variational estimation for $\mathrm{I}(X;Y)$. We start by decomposing $\mathrm{I}(X;Y)$ as follows:

---

**Algorithm 2** Pseudo-code of IGDS

1: **Input:** The number of iterations $N$, $\boldsymbol{y}$, noise levels $\{\tilde{\sigma}\}$, pre-trained encoder $f(\cdot)$ and classifier $g(\cdot)$, $\eta > 0, \beta > 0$, $\tau > 0$, IPC n, target label $y$.
2: $\boldsymbol{x}_N \sim \mathcal{N}(\mathbf{0}, \mathbf{I})$
3: **for** $t = N-1, N-2, \ldots, 0$ **do**
4: $\quad \hat{\boldsymbol{s}} \leftarrow \boldsymbol{s}_\theta(\boldsymbol{x}_t, t)$
5: $\quad \tilde{\boldsymbol{x}}_0 \leftarrow \frac{1}{\sqrt{\bar{\alpha}_t}}(\boldsymbol{x}_t + (1 - \bar{\alpha}_t)\hat{\boldsymbol{s}})$
6: $\quad \boldsymbol{z} \sim \mathcal{N}(\mathbf{0}, \mathbf{I})$.
7: $\quad \boldsymbol{x}'_{t-1} \leftarrow \frac{\sqrt{\alpha_t}(1-\bar{\alpha}_{t-1})}{1-\bar{\alpha}_t}\boldsymbol{x}_t + \frac{\sqrt{\bar{\alpha}_{t-1}}\beta_t}{1-\bar{\alpha}_t}\tilde{\boldsymbol{x}}_0 + \tilde{\sigma}_t\boldsymbol{z}$.
8: $\quad \hat{\boldsymbol{x}}_{t-1} = f(\boldsymbol{x}'_{t-1})$
9: $\quad p_{\hat{\boldsymbol{x}}_{t-1}} = \sigma(\hat{\boldsymbol{x}}_{t-1}/\tau)$
10: $\quad \hat{Q}^y_{t-1} = \frac{1}{n}\sum p_{\hat{\boldsymbol{x}}_{t-1}}$.detach()
11: $\quad \hat{\boldsymbol{y}}_{t-1} = g(\hat{\boldsymbol{x}}_{t-1})$
12: $\quad$ Compute $\hat{\mathcal{L}}_{IGDS}$ using eq. (29)
13: $\quad \boldsymbol{x}_{t-1} \leftarrow \boldsymbol{x}'_{t-1} + \eta \, \nabla_{\boldsymbol{x}_{t-1}} \hat{\mathcal{L}}_{IGDS}$.
14: **end for**
15: **Output:** $\boldsymbol{x}_0$

---

$$\mathrm{I}(X;Y) = \mathrm{I}(\hat{X};Y) + \mathrm{I}(Y;X|\hat{X}). \qquad (3)$$

The first term on the right-hand side of eq. (3), namely $I(\hat{X}; Y)$, is difficult to compute directly. To overcome this difficulty, we introduce the following Proposition:

**Proposition 1.** Consider a linear classifier $g_{\boldsymbol{\psi}}(\hat{X}) = \hat{Y} = \boldsymbol{\psi}\hat{X}$, parameterized by $\boldsymbol{\psi} \in \mathbb{R}^{n \times m}$ with $m \geq n$. If $\boldsymbol{\psi}$ has full column rank, then

$$I(\hat{X}; Y) = I(\hat{Y}; Y). \tag{4}$$

Now, we can write

$$I(\hat{Y}; Y) := H(Y) - H(Y|\hat{Y}) \tag{5}$$

$$\geq H(Y) - H(Y|\hat{Y}, Y) \tag{6}$$

$$= H(Y) + \mathbb{E}_Y \log P_{Y|\hat{Y}}, \tag{7}$$

where the inequality in eq. (6) becomes equality if $P_{Y|\hat{Y}} = P_Y$, i.e., the classifier is Bayes-optimal. Now, the quantities in eq. (7) can be easily computed; specifically, $H(Y)$ is simply the entropy of the ground truth distribution, which is constant for a given dataset., and $\mathbb{E}_Y \log P_{Y|\hat{Y}}$ is the average of cross-entropy of the output, In practice, we use one-hot probabilities to estimate the Bayes probabilities (Ye et al., 2024).

The second term on the right-hand side of eq. (3), namely $I(Y; X|\hat{X})$, is difficult to compute directly. In what follows, we propose to minimize this term so that $I(\hat{X}; Y)$ forms a tight lower bound on $I(X; Y)$. To motivate this, we first present proposition 2.

**Proposition 2.** For an encoder $f_{\boldsymbol{\theta}}(\cdot)$ parametrized by $\boldsymbol{\theta}$

$$\min_{\boldsymbol{\theta}} I(Y; X|\hat{X}) \equiv \max_{\boldsymbol{\theta}} I(X; \hat{X}). \tag{8}$$

As per proposition 2, we shall train $f_{\boldsymbol{\theta}}(\cdot)$ to maximize $I(X; \hat{X})$. The details of this training process are provided in section 4.1.3. In this manner, we find a variational estimation for $I(X; Y)$ which we denote by $\underline{I}(X; Y)$. Particularly,

$$\underline{I}(X; Y) = H(\hat{Y}) + \mathbb{E}_Y \log P_{\hat{Y}|Y}. \tag{9}$$

### 4.1.2 Variational Estimates for $H(X|Y)$

The term $H(X|Y)$ can be expanded as

$$H(X|Y) = I(X; \hat{X}|Y) + H(X|\hat{X}, Y). \tag{10}$$

To compute $I(X; \hat{X}|Y)$, we introduce the following Proposition.

**Proposition 3.** Assume that the feature representation $\hat{X}$ has zero mean (He et al., 2015; Hinton et al., 2015). With $\sigma(\cdot)$ denoting the softmax, it follows that $I(X; \hat{X}|Y) = I(X; \sigma(\hat{X})|Y)$.

Despite $I(X; \hat{X}|Y)$, the term $I(X; \sigma(\hat{X})|Y)$ can indeed be calculated analytically using the same approach as used in (Yang et al., 2025; Ye et al., 2024):

$$I(X; \sigma(\hat{X})|Y) = \sum_{y \in [C]} P_Y(y) \, I(X; \sigma(\hat{X})|y) \tag{11}$$

$$= \sum_{y \in [C]} P_Y(y) \, \mathbb{E}_{X|Y} \mathrm{KL}(\sigma(\hat{X})||Q^y) \tag{12}$$

$$= \mathbb{E}_{X,Y} \mathrm{KL}(\sigma(\hat{X})||Q^Y). \tag{13}$$

Following (Yang et al., 2025), we **define** $Q^y = \frac{1}{|D_y|} \sum_{x \in D_y} \sigma(\hat{X}), \forall y \in [C]$, i.e., the empirical class-conditional mean of $\sigma(\hat{X})$. In addition, the term $H(X|\hat{X}, Y)$ is not easy to compute, so we introduce the following proposition to minimize it such that $I(X; \hat{X}|Y)$ becomes a tight lower bound for $H(X|Y)$.

**Proposition 4.** For an encoder $f_{\boldsymbol{\theta}}(\cdot)$ parametrized by $\theta$

$$\min_{\theta} \mathrm{H}(X|\hat{X}, Y) \equiv \max_{\theta} \mathrm{I}(X; \hat{X}). \tag{14}$$

As such, we have found a variational estimation for $\mathrm{H}(X|Y)$ which we denote by $\underline{\mathrm{H}}(X|Y)$. Particularly,

$$\underline{\mathrm{H}}(X|Y) = \mathbb{E}_{X,Y} \mathrm{KL}(\sigma(\hat{X})||Q^Y). \tag{15}$$

### 4.1.3 Training Variational Estimator

According to proposition 2 and proposition 4, obtaining tight lower bounds for $\mathrm{I}(X; Y)$ and $\mathrm{H}(X|Y)$, denoted as $\underline{\mathrm{I}}(X; Y)$ and $\underline{\mathrm{H}}(X|Y)$ respectively, requires training $f_{\boldsymbol{\theta}}(\cdot)$ to maximize $\mathrm{I}(X; \hat{X})$ and train $g_{\boldsymbol{\psi}}(\cdot)$ with cross-entropy. We refer to this DNN, which consists of the concatenation of $f_{\boldsymbol{\theta}}(\cdot)$ and $g_{\boldsymbol{\psi}}(\cdot)$, as the variational estimator (VE) since it provides tight estimations $\underline{\mathrm{I}}(X; Y)$ and $\underline{\mathrm{H}}(X|Y)$. The training procedure for the VE is described in this subsection. To establish the theoretical foundation for this approach, we first introduce the following theorem.

**Theorem 1.** Consider the mapping $\hat{X} = f_{\boldsymbol{\theta}}(X)$, where $f_{\boldsymbol{\theta}}(\cdot)$ is an encoder parametrized by $\boldsymbol{\theta}$. Then,

$$\max_{\boldsymbol{\theta}} \mathrm{I}(X; \hat{X}) \equiv \min_{\boldsymbol{\theta}} \left[ \mathrm{H}(\tilde{Y}|\hat{X}) - \mathrm{I}(X; \hat{X}|\tilde{Y}) \right], \tag{16}$$

where $\tilde{Y}$ is an auxiliary random variable that denotes the class assignment in the classification task.

*Proof.* Using the chain rule, the mutual information $\mathrm{I}(X; \tilde{Y}, \hat{X})$ can be expanded in two different ways:

$$\mathrm{I}(X; \tilde{Y}, \hat{X}) = \mathrm{I}(X; \tilde{Y}|\hat{X}) + \mathrm{I}(X; \hat{X}) \tag{17a}$$

$$\mathrm{I}(X; \tilde{Y}, \hat{X}) = \mathrm{I}(X; \hat{X}|\tilde{Y}) + \mathrm{I}(X; \tilde{Y}). \tag{17b}$$

By setting the right hand side of eq. (17a) and eq. (17b) equal to each other, we obtain

$$\mathrm{I}(X; \hat{X}) = \mathrm{I}(X; \tilde{Y}) - \mathrm{I}(X; \tilde{Y}|\hat{X}) + \mathrm{I}(X; \hat{X}|\tilde{Y}) \tag{18}$$

$$= \mathrm{I}(X; \hat{X}; \tilde{Y}) + \mathrm{I}(X; \hat{X}|\tilde{Y}) \tag{19}$$

$$= \mathrm{I}(\hat{X}; \tilde{Y}) + \mathrm{I}(X; \hat{X}|\tilde{Y}) \tag{20}$$

$$= \mathrm{H}(\tilde{Y}) - \mathrm{H}(\tilde{Y}|\hat{X}) + \mathrm{I}(X; \hat{X}|\tilde{Y}), \tag{21}$$

where eq. (19) follows from the definition of interaction information, and eq. (20) holds because $\tilde{Y} \to X \to \hat{X}$ forms a Markov chain. $\square$

As per theorem 1, to maximize $\mathrm{I}(X; \hat{X})$ one can instead maximize the three terms: $\mathrm{H}(\tilde{Y})$, $-\mathrm{H}(\tilde{Y}|\hat{X})$, and $\mathrm{I}(X; \hat{X}|\tilde{Y})$. In the following, we discuss how to maximize these three terms.

● $\mathrm{H}(\tilde{Y})$ depends only on the statistics of the random variable $\tilde{Y}$. In addition, since it is an auxiliary random variable, it can be defined such that $\mathrm{H}(\tilde{Y})$ is maximized. The following Proposition establishes a condition under which $\mathrm{H}(\tilde{Y})$ is maximized.

**Proposition 5.** Given a dataset $\mathcal{D}$, the entropy $\mathrm{H}(\tilde{Y})$ is maximized when each individual sample $x \in \mathcal{D}$ is associated with a unique $\tilde{y}$, and the probability distribution of $\mathrm{P}_{\tilde{Y}} = \frac{1}{|\mathcal{D}|}$, $\forall \tilde{y}$.

As per proposition 5, $\mathrm{H}(\tilde{Y})$ is maximized when the task is formulated as an instance discrimination task.

● To maximize $-\mathrm{H}(\tilde{Y}|\hat{X})$, we first note that $\mathrm{H}(\tilde{Y}|\hat{X}) = -\sum P(\hat{x}, y) \log P(y|\hat{x})$. Since we formulate the problem as an instant discrimination task, and following (Gálvez et al., 2023; Oord et al., 2018), we approximate the conditional probability $P(y|\hat{x})$ as

$$P(\tilde{y}_i|\hat{x}_j) \approx \frac{\exp(\langle \hat{x}_i, \hat{x}_j \rangle / \tau)}{\sum_k \exp(\langle \hat{x}_i, \hat{x}_k \rangle / \tau)}, \tag{22}$$

where $\tau$ is a predetermined hyperparameter, usually referred to as the temperature (He et al., 2020). Then,

$$\mathrm{H}(\tilde{Y}|\hat{X}) \le \mathrm{H}(\tilde{Y}|\hat{X}, \hat{Y}|X) = -\mathbb{E}\log\left[\frac{\exp(\langle\hat{x}_i, \hat{x}_i\rangle/\tau)}{\sum_k \exp(\langle\hat{x}_i, \hat{x}_k\rangle/\tau)}\right], \tag{23}$$

where the expectation is take over $P_{\tilde{Y}|\hat{X}}$. The conditional entropy can be minimized by minimizing eq. (23).

• $\mathrm{I}(X; \hat{X}|\tilde{Y})$ can be maximized by iteratively maximizing eq. (13) and updating the $Q^y = \frac{1}{|D_y|}\sum_{x\in D_y}\sigma(\hat{X})$.

**Remark 1.** We acknowledge the similarity between VE and information bottleneck theory (Tishby et al., 2000; Saxe et al., 2018). In section D, we elaborate on this connection and derive the objective function using an alternative approach.

**Remark 2.** The tightness of our variational lower bounds for the contextual and prototype information follows from the decompositions in Equations (3) and (10). As training maximizes $\mathrm{I}(X; \hat{X})$, the corresponding residual terms vanish, and the bounds attain the equalities characterized when $\mathrm{I}(X; \hat{X})$ is maximized, i.e. $\mathrm{I}(X; \hat{X}) = \mathrm{H}(X)$. We empirically validate this tightness in Section 5.7 on a synthetic Gaussian dataset, where the proposed lower bounds closely match the ground truth.

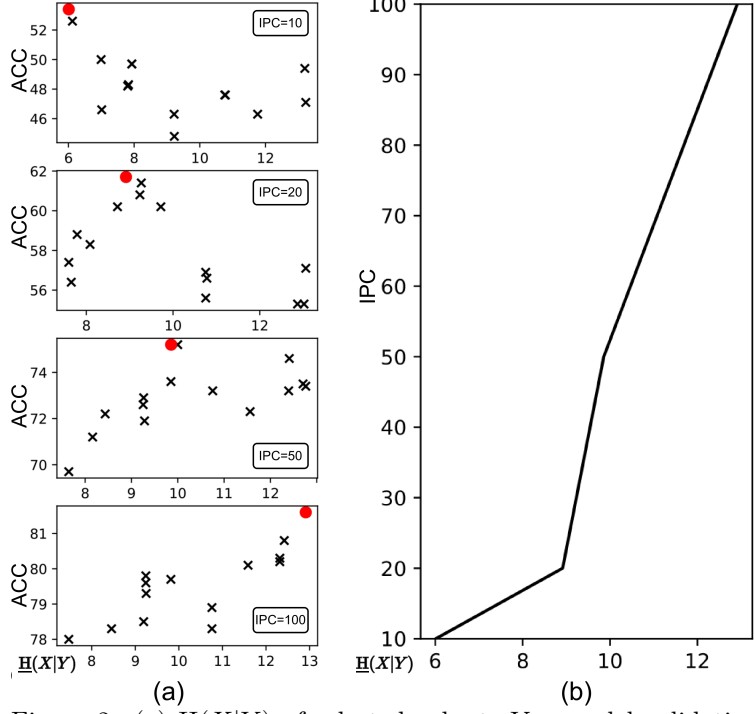

Figure 2: (a) $\underline{\mathrm{H}}(X|Y)$ of selected subsets Vs. model validation accuracy for different IPC settings; the subset associated with the highest model accuracy is marked with a red dot. (b) $\underline{\mathrm{H}}(X|Y)$ of the best subset compared across different IPC settings.

### 4.2 Information-Guided Diffusion Sampling

Based on the discussion above, to maximize $\mathrm{I}(X; \hat{X})$ for an encoder $f_{\boldsymbol{\theta}}(\cdot)$, one can train it to maximize the following objective function:

$$\mathcal{J}_{\mathrm{VE}} = -\mathbb{E}_{\hat{X}, Y}\log\left[\frac{\exp(\langle\hat{x}_i, \hat{x}_i\rangle/\tau)}{\sum_k \exp(\langle\hat{x}_i, \hat{x}_k\rangle/\tau)}\right] + \lambda\,\mathbb{E}_{\hat{X}, Y}\mathrm{KL}(\sigma(\hat{X})||Q^Y), \tag{24}$$

where $\lambda$ is a hyperparameter that balances the effects of two terms in the objective function. We note that VE reduces to MoCo (He et al., 2020) when $\lambda = 0$.

By maximizing $\mathcal{J}_{\mathrm{VE}}$, we can effectively train $f_{\boldsymbol{\theta}}(\cdot)$. The pseudo-code for this training procedure is provided in algorithm 1.

Once $f_{\boldsymbol{\theta}}(\cdot)$ is trained, we proceed to the next step. We freeze the parameters of $f_{\boldsymbol{\theta}}(\cdot)$ and train the classifier $g_{\boldsymbol{\psi}}(\cdot)$ using the standard cross-entropy (CE) loss. Once $f_{\boldsymbol{\theta}}(\cdot)$ and $g_{\boldsymbol{\psi}}(\cdot)$ are trained, they are fixed and used during the sampling process of the diffusion model as discussed in the following paragraph.

We use DDPM as an image prior. At reverse step $t \to t-1$, given $x_t$, a learned score function $s_\theta(x_t, t) \approx \nabla_{x_t}\log p_t(x_t)$ yields an estimate of the clean sample

$$\tilde{x}_0 = \frac{1}{\sqrt{\bar{\alpha}_t}}\Big(x_t + (1 - \bar{\alpha}_t)\,s_\theta(x_t, t)\Big). \tag{25}$$

Conditioned on $\tilde{x}_0$, we sample the previous state from the DDPM posterior $q(x_{t-1} \mid x_t, \tilde{x}_0)$:

$$x'_{t-1} = \frac{\sqrt{\alpha_t}\left(1 - \bar{\alpha}_{t-1}\right)}{1 - \bar{\alpha}_t}\, x_t \; + \; \frac{\sqrt{\bar{\alpha}_{t-1}}\,\beta_t}{1 - \bar{\alpha}_t}\, \tilde{x}_0 \; + \; \tilde{\sigma}_t\, z, \qquad z \sim \mathcal{N}(0, I), \tag{26}$$

$$\tilde{\sigma}_t^2 = \frac{1 - \bar{\alpha}_{t-1}}{1 - \bar{\alpha}_t}\, \beta_t. \tag{27}$$

Here $\beta_t$ is the noise schedule, $\alpha_t := 1 - \beta_t$, and $\bar{\alpha}_t := \prod_{j=1}^{t} \alpha_j$. To recap, our primary objective was to maximize the function $\mathrm{I}(X;Y) + \beta\,\mathrm{H}(X|Y)$ (see eq. (2)) during the sampling process of the diffusion model. To achieve this, we derived variational estimates for $\mathrm{I}(X;Y)$ (eq. (9)) and $\mathrm{H}(X|Y)$ (eq. (15)) by leveraging the training of a variational estimator (VE). Using these estimates, the objective function in eq. (2) is reformulated as:

$$\mathcal{L}_{IGDS} = \mathbb{E} \log P_{Y|\hat{Y}} + \beta\, \mathrm{I}(\hat{X}; \sigma(\hat{X})|Y). \tag{28}$$

Given $n$ IPC, for class $y$ we form a mini-batch $|B_y|$ of the size $|B_y| = n$ and approximate $\mathcal{L}_{IGDS}$ at reverse step $t-1$ by

$$\hat{\mathcal{L}}_{IGDS} = \frac{1}{n} \sum_{x'_{t-1} \in B_y} \log P\big(y|g(f(x'_{t-1}))\big) + \beta\, \mathrm{KL}(\sigma(f(x'_{t-1})/\tau)\|\hat{Q}^y_{t-1}), \tag{29}$$

with the batch statistic

$$\hat{Q}^y_{t-1} := \frac{1}{n} \sum_{x'_{t-1} \in B_y} \sigma(f(x'_{t-1})/\tau). \tag{30}$$

During sampling, we maximized empirical objective $\hat{\mathcal{L}}_{IGDS}$[2]. After each reverse-diffusion step we apply

$$x_{t-1} \leftarrow x'_{t-1} + \eta \nabla_{\boldsymbol{x}_{t-1}}\, \hat{\mathcal{L}}_{IGDS}, \tag{31}$$

where $\eta$ is the pre-iteration step size. Consistent with our experimental implementation, algorithm 2 illustrates how our method can be integrated with the denoising diffusion probabilistic model (DDPM) (Ho et al., 2020), specifically, line 13 uses the gradient to guide the diffusion step. We refer to the resulting DM-based sampling method as information-guided diffusion sampling (IGDS).

Table 1: Comparing the model's performance in terms of accuracy on the ImageWoof validation set. All results are evaluated at a resolution of $256 \times 256$. We use **bold** number and asterisk (*) to denote the best and the second best results, respectively.

| IPC (Ratio) | Test Model | Random | K-Center | Herding | DiT | DM | IDC-1 | GLaD | MiniMax | RDED | Ours |
|---|---|---|---|---|---|---|---|---|---|---|---|
| 1 (0.08%) | ConvNet-6 | $14.2_{\pm0.9}$ | $15.6_{\pm1.0}$ | - | $12.7_{\pm0.6}$ | $21.1_{\pm0.5}{}^*$ | - | - | $15.2_{\pm0.6}$ | $18.5_{\pm0.9}$ | $\mathbf{23.1}_{\pm0.8}$ |
| | ResNetAP-10 | $17.8_{\pm2.4}$ | $18.3_{\pm0.6}$ | - | $18.0_{\pm1.3}$ | - | - | - | $18.9_{\pm2.4}{}^*$ | - | $\mathbf{23.6}_{\pm0.9}$ |
| | ResNet-18 | $13.5_{\pm0.4}$ | $12.5_{\pm0.8}$ | - | $15.3_{\pm0.7}$ | - | - | - | $14.6_{\pm0.6}$ | $20.8_{\pm1.2}{}^*$ | $\mathbf{22.8}_{\pm0.8}$ |
| 10 (0.4%) | ConvNet-6 | $24.3_{\pm1.1}$ | $19.4_{\pm0.9}$ | $26.7_{\pm0.5}$ | $34.2_{\pm1.1}$ | $26.9_{\pm1.2}$ | $33.3_{\pm1.1}$ | $33.8_{\pm0.9}$ | $37.0_{\pm1.0}$ | $40.6_{\pm2.0}{}^*$ | $\mathbf{41.9}_{\pm1.5}$ |
| | ResNetAP-10 | $29.4_{\pm0.8}$ | $22.1_{\pm0.1}$ | $32.0_{\pm0.3}$ | $34.7_{\pm0.5}$ | $30.3_{\pm1.2}$ | $39.1_{\pm0.5}$ | $32.9_{\pm0.9}$ | $39.2_{\pm1.3}{}^*$ | - | $\mathbf{43.5}_{\pm0.3}$ |
| | ResNet-18 | $27.7_{\pm0.9}$ | $21.1_{\pm0.4}$ | $30.2_{\pm1.2}$ | $34.7_{\pm0.4}$ | $33.4_{\pm0.7}$ | $37.3_{\pm0.2}$ | $31.7_{\pm0.8}$ | $37.6_{\pm0.9}$ | $38.5_{\pm2.1}{}^*$ | $\mathbf{40.7}_{\pm0.5}$ |
| 20 (1.6%) | ConvNet-6 | $29.1_{\pm0.7}$ | $21.5_{\pm0.8}$ | $29.5_{\pm0.3}$ | $36.1_{\pm0.8}$ | $29.9_{\pm1.0}$ | $35.5_{\pm0.8}$ | - | $37.6_{\pm0.2}{}^*$ | - | $\mathbf{45.7}_{\pm0.6}$ |
| | ResNetAP-10 | $32.7_{\pm0.4}$ | $25.1_{\pm0.7}$ | $34.9_{\pm0.1}$ | $41.1_{\pm0.8}$ | $35.2_{\pm0.6}$ | $43.3_{\pm0.3}$ | - | $45.8_{\pm0.5}{}^*$ | - | $\mathbf{55.1}_{\pm0.6}$ |
| | ResNet-18 | $29.7_{\pm0.5}$ | $23.6_{\pm0.3}$ | $32.2_{\pm0.6}$ | $40.5_{\pm0.5}$ | $29.8_{\pm1.7}$ | $38.6_{\pm0.2}$ | - | $42.5_{\pm0.6}{}^*$ | - | $\mathbf{49.9}_{\pm0.7}$ |
| 50 (3.8%) | ConvNet-6 | $41.3_{\pm0.6}$ | $36.5_{\pm1.0}$ | $40.3_{\pm0.7}$ | $46.5_{\pm0.8}$ | $44.4_{\pm1.0}$ | $43.9_{\pm1.2}$ | - | $53.9_{\pm0.6}$ | $61.5_{\pm0.3}{}^*$ | $\mathbf{65.3}_{\pm1.4}$ |
| | ResNetAP-10 | $47.2_{\pm1.3}$ | $40.6_{\pm0.4}$ | $49.1_{\pm0.7}$ | $49.3_{\pm0.2}$ | $47.1_{\pm1.1}$ | $48.3_{\pm1.0}$ | - | $56.3_{\pm1.0}{}^*$ | - | $\mathbf{70.2}_{\pm0.8}$ |
| | ResNet-18 | $47.9_{\pm1.8}$ | $39.6_{\pm1.0}$ | $48.3_{\pm1.2}$ | $50.1_{\pm0.5}$ | $46.2_{\pm0.6}$ | $48.3_{\pm0.8}$ | - | $57.1_{\pm0.6}$ | $68.5_{\pm0.7}{}^*$ | $\mathbf{71.3}_{\pm0.2}$ |
| 100 (7.7%) | ConvNet-6 | $52.2_{\pm0.4}$ | $45.1_{\pm0.5}$ | $54.4_{\pm1.1}$ | $53.4_{\pm0.3}$ | $55.0_{\pm1.3}$ | $53.2_{\pm0.9}$ | - | $61.1_{\pm0.7}{}^*$ | - | $\mathbf{67.2}_{\pm0.2}$ |
| | ResNetAP-10 | $59.4_{\pm1.0}$ | $54.8_{\pm0.2}$ | $61.7_{\pm0.9}$ | $58.3_{\pm0.8}$ | $56.4_{\pm0.8}$ | $56.1_{\pm0.9}$ | - | $64.5_{\pm0.2}{}^*$ | - | $\mathbf{76.7}_{\pm0.3}$ |
| | ResNet-18 | $61.5_{\pm1.3}$ | $50.4_{\pm0.4}$ | $59.3_{\pm0.7}$ | $58.9_{\pm1.3}$ | $60.2_{\pm1.0}$ | $58.3_{\pm1.2}$ | - | $65.7_{\pm0.4}{}^*$ | - | $\mathbf{77.3}_{\pm0.7}$ |

---

[2]In practice, guidance employs the empirical mini-batch objective in eq. (29). At each reverse step, the gradient is computed with respect to each sample, ensuring that the per-sample update serves as a well-defined estimator of the population gradient.

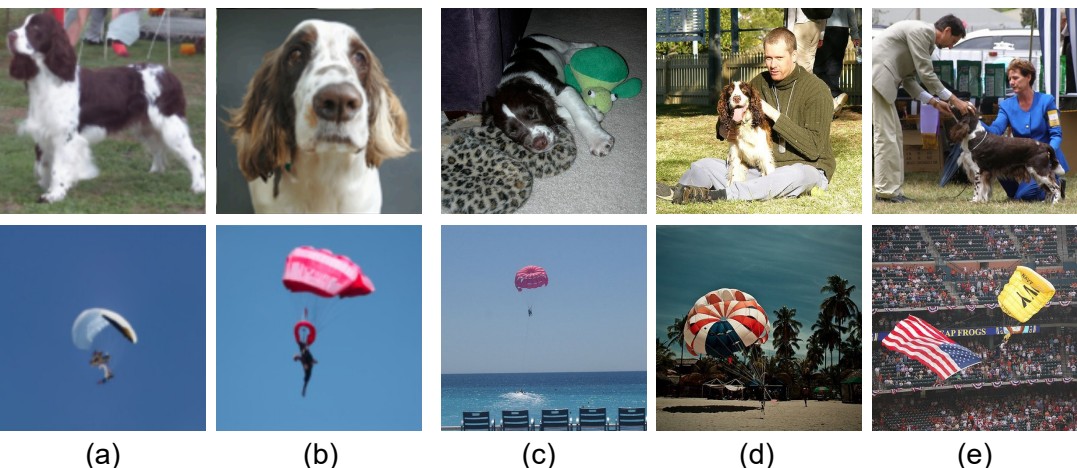

|  |  |  |  |  |
|---|---|---|---|---|
| (a) | (b) | (c) | (d) | (e) |

Figure 3: Illustration of prototype and contextual information. (*i*) Top: English Springer; (*ii*) Bottom: Parachute. The first two columns show synthetic images with low contextual information, while the last three columns display natural images from the same classes.

Table 2: Comparing the model's performance in terms of accuracy on the ImageNette validation set. All results are evaluated at a resolution of $256 \times 256$. We use **bold** number and asterisk (*) to denote the best and the second best results, respectively.

| Model | ResNetAP-10 | | | | | ResNet-18 | | |
|---|---|---|---|---|---|---|---|---|
| IPC | Random | DiT | DM | MiniMax | Ours | RDED | $SRe^2L$ | Ours |
| 1 | $26.7_{\pm 1.0}$ | $27.3_{\pm 0.9}$ | - | $30.5_{\pm 0.8}{}^{*}$ | $\mathbf{39.6_{\pm 1.3}}$ | $35.8_{\pm 1.0}{}^{*}$ | $19.1_{\pm 1.1}$ | $\mathbf{35.9_{\pm 0.7}}$ |
| 10 | $54.3_{\pm 1.6}$ | $59.1_{\pm 0.7}$ | $60.8_{\pm 0.6}$ | $62.0_{\pm 0.2}{}^{*}$ | $\mathbf{68.3_{\pm 0.2}}$ | $61.4_{\pm 0.4}{}^{*}$ | $29.4_{\pm 3.0}$ | $\mathbf{64.3_{\pm 0.6}}$ |
| 20 | $63.5_{\pm 0.5}$ | $64.8_{\pm 1.2}$ | $66.5_{\pm 1.1}$ | $66.8_{\pm 0.4}{}^{*}$ | $\mathbf{72.4_{\pm 0.7}}$ | - | - | $\mathbf{70.9_{\pm 0.3}}$ |
| 50 | $76.1_{\pm 1.1}$ | $73.3_{\pm 0.9}$ | $76.2_{\pm 0.4}$ | $76.6_{\pm 0.2}{}^{*}$ | $\mathbf{81.0_{\pm 0.5}}$ | $80.4_{\pm 0.5}{}^{*}$ | $40.9_{\pm 0.3}$ | $\mathbf{81.2_{\pm 0.4}}$ |

## 5 Experiments

### 5.1 Experimental Setup

• **Implementation Details of IGDS.** We adopt the pre-trained DDPM model (Dhariwal & Nichol, 2021) and use the pre-trained MoCo model (He et al., 2020) as the encoder. To smooth gradients, we follow (Ma et al., 2024) and replace the ReLU activation function with SoftPlus (Dugas et al., 2000) using $\alpha = 3$. A linear classifier is then trained on top of the frozen encoder. During the DM sampling, we set the temperature to $\tau = 0.07$ and run the diffusion process for 250 steps in all experiments. Following (Sun et al., 2024), we enhance sample information by merging four images from the same class into a single composite image. We report the evaluation protocol in section F. Additionally, section H provides sample images from the distilled datasets generated by IGDS. All experiments are conducted on a single NVIDIA V100 GPU. Full implementation details, including code and configurations, are available in our *GitHub* repository.

• **Datasets.** To evaluate the effectiveness of the proposed method, we conduct experiments mainly on several benchmark datasets. We select ImageNet-1K (Deng et al., 2009) and three well-known subsets of ImageNet: ImageNette, ImageWoof, and Tiny ImageNet (Le & Yang, 2015). ImageNet is a large-scale visual recognition dataset containing approximately 1.2 million training images and 50,000 validation images. ImageNette, a subset of ImageNet, provides a smaller and more manageable dataset for testing deep learning models, while ImageWoof focuses on 10 dog breeds, offering a fine-grained classification task. Both ImageNette and ImageWoof use a spatial resolution of $224 \times 224$. Tiny ImageNet, on the other hand, is a small, balanced subset of ImageNet, with its training set consisting of 200 classes—each class containing 500 samples resized to a spatial resolution of $64 \times 64$.

• **Network Architectures.** Following previous work (Cazenavette et al., 2022b; Cui et al., 2023; Guo et al., 2024), we use ConvNet-4 (LeCun et al., 1998) for the Tiny ImageNet dataset and ConvNet-6 for the ImageWoof dataset. Additionally, we employ ResNetAP-10, a variant of ResNet-10 where all pooling layers are replaced with average pooling, and ResNet-18 for all experiments.

## 5.2 How to select $\beta$ in eq. (28)

As discussed in section 4, the parameter $\beta$ in eq. (28) should be selected based on the IPC. To illustrate this, we generate multiple subsets of Tiny ImageNet with varying $\underline{H}(X|Y)$ and IPC values. To control $\underline{H}(X|Y)$, we apply a weighted sampling method, which is detailed in the Supplementary Materials. We then train ConvNet-4 on these subsets and report the classification accuracies. Finally, we plot the relationship between $\underline{H}(X|Y)$ and model validation accuracy across different IPC settings in fig. 2. As observed, higher IPC settings require greater contextual detail, while lower IPC settings benefit from a stronger emphasis on prototype information. We next visualize prototype and contextual signals to make their semantic meaning explicit.

## 5.3 Semantic Meaning of Prototype and Contextual Information

To better understand the semantic meaning of prototype and contextual information, we visualize samples generated by IGDS with $\beta = 0$, where *only* prototype information is maximized during the DM sampling process. These synthetic im-

Table 3: Comparing the model's performance in terms of accuracy on the Tiny ImageNet validation set. All results are evaluated at a resolution of $64 \times 64$. We use **bold** number and asterisk (*) to denote the best and the second best results, respectively.

| Model | ConvNet-4 | | | |
|---|---|---|---|---|
| IPC | Random | IDM (Zhao et al., 2023) | RDED (Sun et al., 2024) | Ours |
| 1 | $6.7_{\pm 0.4}$ | $10.1_{\pm 0.2}$ | $\mathbf{12.0}_{\pm \mathbf{0.1}}$ | $11.9_{\pm 0.3}{}^{*}$ |
| 10 | $17.6_{\pm 0.3}$ | $21.9_{\pm 0.3}$ | $39.6_{\pm 0.1}{}^{*}$ | $\mathbf{40.7}_{\pm \mathbf{0.3}}$ |
| 50 | $22.4_{\pm 0.2}$ | $27.7_{\pm 0.3}$ | $47.6_{\pm 0.2}{}^{*}$ | $\mathbf{50.3}_{\pm \mathbf{0.2}}$ |
| Model | ResNet-18 | | | |
| IPC | Random | $SRe^2L$ (Yin et al., 2023) | RDED (Sun et al., 2024) | Ours |
| 1 | $2.2_{\pm 0.4}$ | $2.6_{\pm 0.1}$ | $9.7_{\pm 0.4}{}^{*}$ | $\mathbf{9.8}_{\pm \mathbf{0.4}}$ |
| 10 | $14.6_{\pm 0.2}$ | $16.1_{\pm 0.2}$ | $\mathbf{41.9}_{\pm \mathbf{0.2}}$ | $41.2_{\pm 0.1}{}^{*}$ |
| 50 | $35.6_{\pm 0.3}$ | $41.1_{\pm 0.4}$ | $58.2_{\pm 0.1}{}^{*}$ | $\mathbf{60.1}_{\pm \mathbf{0.5}}$ |

Table 4: Performance comparison over ResNet-18 on CIFAR-10.

| IPC | Test Model | $SRe^2L$ | RDED | DIT-IGD | Ours |
|---|---|---|---|---|---|
| 10 | ResNet-18 | 29.3 | **37.1** | 35.8 | 37.0 |
| 50 | ResNet-18 | 45.0 | 62.1 | 63.5 | **64.9** |

Table 5: Cross-architecture performance.

| Method | ResNet-101 | MobileNet-V2 | EfficientNet-B0 | SwinT |
|---|---|---|---|---|
| MiniMax-IGD | 53.4 | 39.7 | **48.5** | 44.8 |
| MiniMax-ours | **53.6** | **39.9** | 48.3 | **45.9** |

ages are shown in the first two columns of fig. 3 (columns (a) and (b)). For direct contrast, three natural images randomly selected from the same class are displayed in the last three columns (columns (c) to (e)). As observed, the synthetic images with low contextual information feature plain backgrounds and minimal context, whereas the natural images exhibit richer contextual details.

## 5.4 Comparison with State-of-the-art Methods

We first report the experimental results for ImageWoof in table 1. As seen, at a low IPC setting (IPC-1), the performance of all generative-based dataset distillation methods, except IGDS, is close to that of a randomly selected subset. However, as the IPC increases, the performance gap between the baseline methods and the random selection also increases. Nevertheless, none of the baseline methods outperforms IGDS.

We also present the experimental results for ImageNette and Tiny ImageNet in table 2 and table 3, respectively. The results follow a similar trend to those on the ImageWoof dataset.

To assess IGDS in a low-resolution setting, we benchmark it on CIFAR-10 with a ResNet-18 alongside strong baselines (Table 4). IGDS matches or surpasses prior state-of-the-art performance on this dataset. We defer the transfer learning results to the appendix G.

### 5.5 Cross-architecture performance

To evaluate the robustness of our distilled datasets across diverse model families, we follow the IGD protocol and measure performance when training four different architectures on the same distilled coresets. table 5 summarizes the test accuracies achieved by MiniMax-IGD and our method under the IPC-10 setting.

### 5.6 Combining with Priors Beyond DDPM

An important advantage of our approach is its flexibility to integrate with a wide range of generative priors beyond DDPM. To illustrate this capability, we conducted experiments combining our method with the MiniMax-DIT prior, which is recognized as a more advanced prior often leading to stronger performance. As shown in table 6, when using the MiniMax-DIT prior, our method achieves an accuracy of 48.6% on the ImageWoof dataset under IPC-10. This result demonstrates that our approach not only remains effective when paired with stronger priors but can also be seamlessly combined with alternative generative models to further enhance performance.

### 5.7 Tightness of the Variational Lower Bounds

As the true distribution of the natural image is inaccessible, we validate our variational estimator on a toy dataset to test whether it yields a tighter lower bound than cross-entropy training. Specifically, the dataset has three classes uniformly distributed, each data point is five-dimension, and, for each class, samples are drawn from a normal distribution with identity covariance. The three class means form an equilateral triangle, we define the inter-class distance as

$$\delta = \|\mu_c - \mu'_c\|_{c \neq c'}, \qquad (32)$$

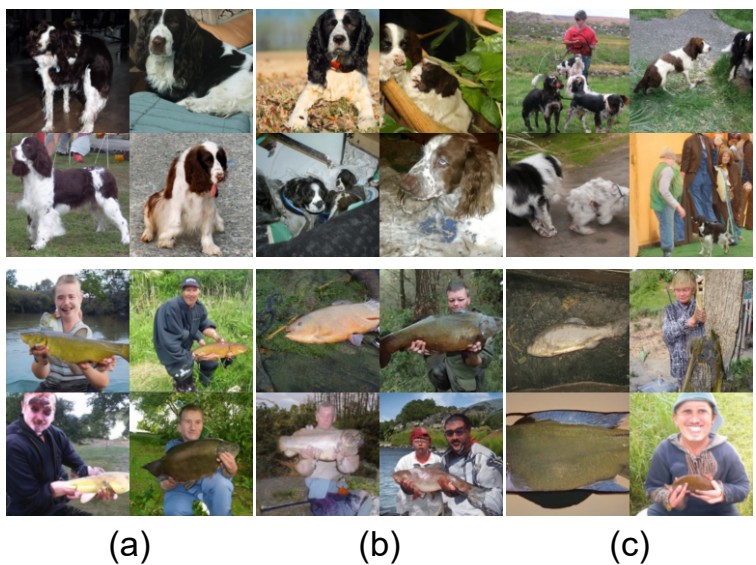

(a)          (b)          (c)

Figure 4: Generated images of two classes: English Springer (first row) and Tench (second row), with varying $\beta$ values. Columns (a), (b), and (c) correspond to $\beta = \{0, 0.1, 0.5\}$, respectively.

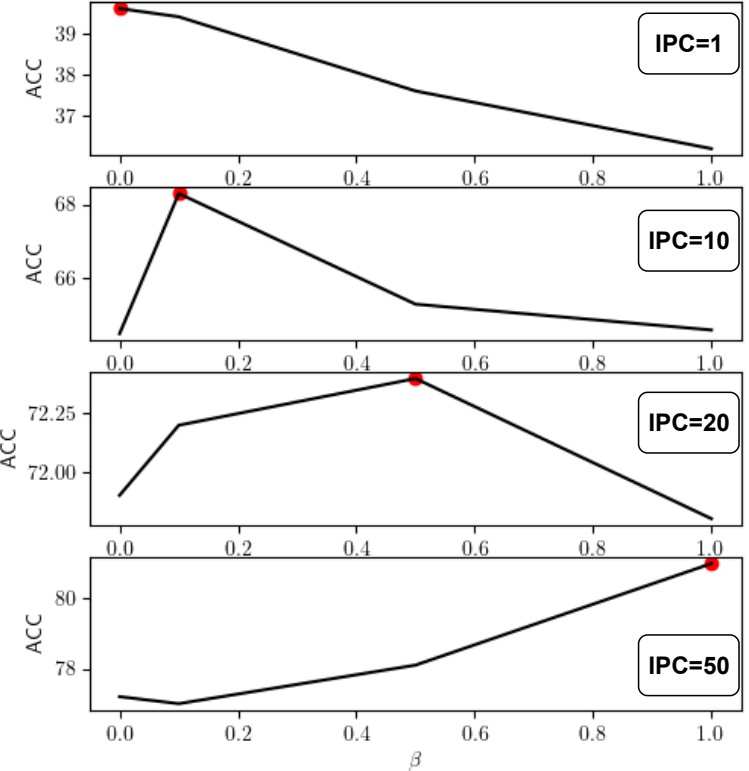

Figure 5: The model's accuracy on the distilled dataset Vs. $\beta$. As observed, lower IPC settings favor smaller $\beta$ values, whereas higher IPC settings require an increased $\beta$ value accordingly.

and scan the inter-class distance in $\{0.5, 1, 1.5, 2, 3, 4\}$. Since $X|Y \sim \mathcal{N}(\mu_c|\Sigma_c)$ the contextual information is

$$\mathrm{H}(X|Y) = \sum_c \pi_c \frac{1}{2} \log((2\pi e)^d \det(\Sigma_c)) \tag{33}$$

$$= \frac{d}{2} \log(2\pi e), \tag{34}$$

where $d$ is the data dimension and $\Sigma_c$ is a identity matrix. We approximate the true $I(X;Y)$ via Monte Carlo by drawing samples from the dataset and computing

$$\mathrm{I}(X;Y) = \mathbb{E}_{p_{(x,y)}}\Big[\log \frac{p(y|x)}{\pi_y}\Big]; \tag{35}$$

$$p(y|x) = \frac{\pi_y \mathcal{N}(x|\mu_y, \Sigma_y)}{\sum_k \pi_k \mathcal{N}(x|\mu_k, \Sigma_k)}. \tag{36}$$

We then generate a dataset of $10^6$ samples and train a three-layer multilayer perceptron with SGD (learning rate 0.01) for 100 epochs. We train two variants—one with cross-entropy loss and one with our proposed variational estimator—and then compute the resulting variational lower bounds of $I(X;Y)$ and $H(X|Y)$. The results are plot in Figure 6.

As observed, the model trained with our variational estimator consistently yields more accurate estimates of contextual information ($H(X|Y)$) than its cross-entropy–trained counterpart, while achieving comparable performance when estimating prototype information ($I(X;Y)$). In the next section, we ablate the IGDS parameter $\beta$ and analyze its impact on the distilled images.

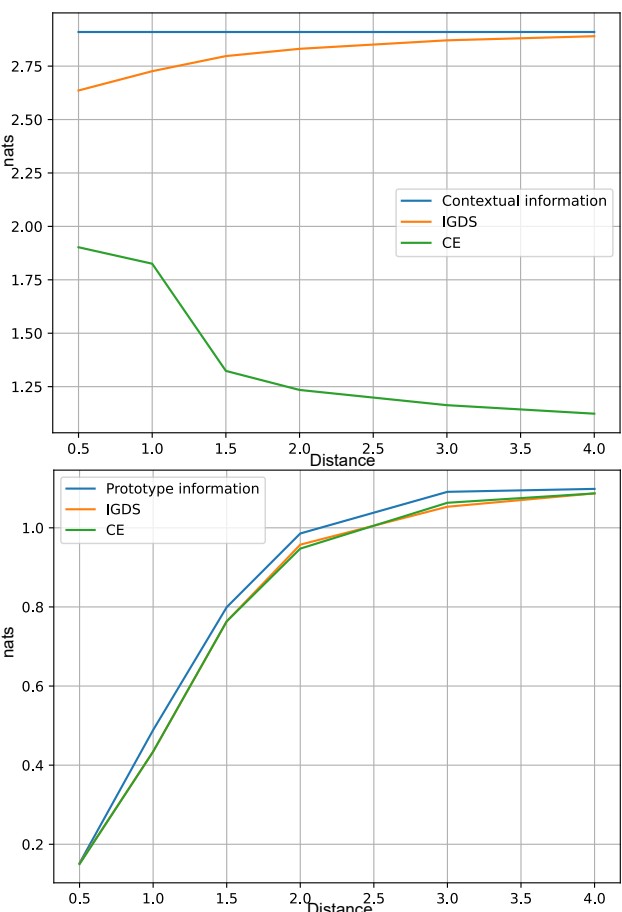

Figure 6: Variational estimation on contextual and prototype information using the IGDS and on the syntactic Gaussian dataset.

## 6 Ablation Study on $\beta$ in IGDS

In this section, we study the impact of $\beta$ on IGDS, the performance of the distilled dataset under different IPC settings, and its effect on the generated images. To this end, we first examine how $\beta$ influences the semantic meaning of generated images, as shown in fig. 4. Specifically, we generate 24 synthetic images for the classes English Springer and Tench, displayed in the first and second rows of fig. 4, respectively. Columns (a), (b), and (c) correspond to $\beta = \{0, 0.1, 0.5\}$. When $\beta = 0$, the generated images contain minimal contextual information. For example, the English Springer images primarily depict the dog itself, while the Tench images consistently depict a person holding the fish. As $\beta$ increases, more contextual elements are incorporated into the generated images, leading to a greater diversity in semantic meaning. This effect is particularly noticeable in the English Springer images, where the background becomes richer compared to those generated with $\beta = 0$. A similar trend can be observed in the Tench images, where additional contextual details emerge as $\beta$ increases. More images generated by IGDS with different $\beta$ values are presented in section H.

The optimal value of $\beta$ should be empirically determined for different IPC settings. To illustrate this, fig. 5 shows the test accuracy of the model as a function of $\beta$ under varying IPC values. As observed, higher IPC settings benefit from a larger $\beta$, aligning with the findings discussed in section 5.2.

Table 6: IGD vs. ours when using MiniMax-DiT as prior.

| IPC | 1 | 10 | 50 | 100 |
|---|---|---|---|---|
| MiniMax-IGD | - | 47.2 | 65.0 | 71.5 |
| MiniMax-Ours | 37.6 | **48.6** | **65.6** | **75.3** |

## 7  Conclusions and Future Work

In this work, we addressed the limitations of diffusion model-based dataset distillation in low-IPC settings through an information-theoretic approach. We identified prototype information $I(X;Y)$ and contextual information $H(X|Y)$ as essential components and proposed maximizing $I(X;Y) + \beta H(X|Y)$ during sampling, with $\beta$ adapted to IPC. To handle intractability, we introduced variational estimations using a deep neural network. Our proposed method, information-guided diffusion sampling (IGDS), seamlessly integrated with diffusion models and achieved state-of-the-art performance on Tiny ImageNet and ImageNet subsets, particularly in low-IPC regimes.

Despite the theoretical contributions and promising results of the proposed information-guided diffusion sampling (IGDS) method, this work has several limitations. First, like previous studies, we use a pretrained diffusion model as the prior distribution for natural images. While this approach is intuitive, its optimality for dataset distillation remains unverified. In addition, it restricts applicability by requiring a pretrained diffusion model for the target dataset. Second, during the IGDS process, gradients must be backpropagated through both the classifier and encoder to guide the diffusion process, increasing computational costs. We report the runtime analysis in section E. Furthermore, our guidance relies on an empirical mini-batch estimator. Under very low IPC $n$, the estimator has high variance (variance increases as $n$ decreases) and yields incomplete mode coverage of the data distribution. In addition, because the method is grounded in NCMI Yang et al. (2025; 2024), it is currently limited to classification. Extending it to other settings—such as regression—will require revisiting the underlying theory. Addressing these limitations is a direction for future work.

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

# Appendix

## A   Weighed Sampling to Generate Subsets with Different $\underline{H}(X|Y)$ values

In this section, we describe how to perform weighted sampling to generate the subset of a dataset with different $\underline{H}(X|Y)$ values. Given a dataset $\mathcal{D}$ of size $n$ with $C$ classes, $\mathcal{D} = \{(\boldsymbol{x}_i, y_i)\}_{i=1}^n$, where each $\boldsymbol{x}_i \in \mathbb{R}^d$ and $y_i \in [C]$, a pretrained encoder $f(\cdot)$ trained using the VE method on $\mathcal{D}$, and a classifier $g(\cdot)$, we first filter out all misclassified samples to ensure that the remaining samples' contextual information can be captured by $g(f(\cdot))$. In the Shannon sense, the contextual information for each sample within class $y$ is quantified using the KL divergence $\mathrm{KL}(\sigma(x)||Q^y)$, where $\sigma(x)$ is a probability vector obtained by applying the softmax function to the feature map $\hat{x}$, and $Q^y$ is estimated as $\frac{1}{|D_y|} \sum_{x \in D_y} \sigma(x)$. Given a target $\alpha_{\underline{H}(\cdot|y)}$ value, we compute the probability $\tilde{P}_{X|Y}(\cdot|y)$ of each sample being selected as follows:

$$\tilde{P}_{X|Y}(x|y) = \frac{\exp(-(\mathrm{KL}(\sigma(x)||Q^y) - \alpha_{\underline{H}(\cdot|y)})^2)}{\sum_{x' \in D^y} \exp(-(\mathrm{KL}(\sigma(x')||Q^y) - \alpha_{\underline{H}(\cdot|y)})^2)}, \tag{37}$$

then, IPC samples are drawn from each class according to the probability $\tilde{P}_{X|Y}(\cdot|y)$. We visualize the samples that map to $Q^Y$ using the pretrained encoder $f(\cdot)$ in Figure 3 to enhance understanding of the semantic meaning of prototype information and contextual information.

## B   Related Work on Neural Information Theory

Information theory offers a unifying lens for representation learning, regularization, and compression in neural networks. The Information Bottleneck (IB) Tishby et al. (2000) formalizes learning as a trade-off between compressing inputs and preserving task-relevant information; the original IB method introduced this principle for stochastic encoders and decoders. Building on IB, Deep Variational Information Bottleneck (VIB) makes the objective trainable with amortized variational inference and the reparameterization trick, demonstrating benefits for generalization and robustness. Alemi et al. (2016)

Estimating or bounding mutual information (MI) reliably at scale is central to many modern objectives. Mutual Information Neural Estimation (MINE) uses the Donsker–Varadhan representation with a learned critic, enabling gradient-based MI estimation in high dimensions Ishmael Belghazi et al. (2018).

One interesting research field of neural information theory is neural compression research operationalizes these ideas directly under rate–distortion objectives. Early end-to-end optimized image compression with analysis/synthesis transforms framed training as RD optimization and highlighted the VAE connection Ballé et al. (2017). Variational image compression with a scale hyperprior introduced a learned hyperprior for accurate entropy models (side-information), improving RD performance Ballé et al. (2018).

Collectively, these threads position information theory not only as an explanatory framework (e.g., IB, information plane) but as a practical toolkit for training objectives, estimators, and codecs that trade off rate, distortion, and task sufficiency.

In contrast to prior work, we train a variational estimator to obtain tighter lower bounds of the prototype information $I(X;Y)$ and contextual information $H(X|Y)$.

## C   Proof of Propositions

### C.1   Proof of proposition 1

*Proof.* We first prove that for any injective function $f$,

$$\mathrm{I}(X;Y) = \mathrm{I}(f(X);Y), \tag{38}$$

To do so, we begin by expanding the mutual information $I(\cdot; \cdot)$, and introducing the variable $Z = f(X)$:

$$I(X; Y) - I(f(X); Y) = H(Y|f(X)) - H(Y|X) \tag{39}$$
$$= H(X|f(X)). \tag{40}$$

Since $f$ is injective, for any output $z = f(X)$, there exists a unique $x$ such that $f(x) = z$. Therefore,

$$P(X = x|f(X) = z) = \begin{cases} 1 & \text{if } z = f(x), \\ 0 & \text{otherwise.} \end{cases} \tag{41}$$

The conditional entropy is then given by:

$$H(X|f(X)) = \mathbb{E}_{f(X)}[H(X|f(X) = z)] \tag{42}$$
$$= \mathbb{E}_{f(X)}0 \tag{43}$$
$$= 0. \tag{44}$$

Thus, if $f$ is injective, we conclude that $I(Y, f(X)) = I(Y, X)$.

Next, we show that for a matrix $\theta \in \mathbb{R}^{m,n}, m \geq n$, if $\theta$ has full column rank, then the linear mapping $\{\hat{X} \to \hat{Y}, \text{ where } \hat{Y} = \theta\hat{X}\}$, is injective.

A function is injective if:

$$\theta x_1 = \theta x_2 \Rightarrow x_1 = x_2, \tag{45}$$

Rearranging, introducing $v = x_1 - x_2$ we get: $\theta v = 0$. For injectivity, we must show that the only solution to $\theta v = 0$ is $v = 0$.

The set of all solutions to $\theta v = 0$ is the null space of $\theta$, denoted as:

$$Null(\theta) = \{v \in \mathbb{R}^n | \theta v = 0\}. \tag{46}$$

If the linear mapping is injective, the only vector in the null space must be the zero vector, *i.e.*, $Null(\theta) = \{0\}$, which means $\theta$ has full column rank. $\square$

We verify that the classifier's matrix after training has full column rank.

### C.2 Proof of proposition 2

$\min_\theta I(Y; X|\hat{X}) \equiv \max_\theta I(X; \hat{X})$

*Proof.*

$$I(Y; X|\hat{X}) = I(Y; X|\hat{X}) \tag{47}$$
$$= H(X|\hat{X}) - H(X|Y, \hat{X}) \tag{48}$$
$$= H(X) - H(X|Y) + H(X|\hat{X}) - H(X) \tag{49}$$
$$= I(X; Y) - I(X; \hat{X}), \tag{50}$$

where $I(X; Y)$ is a constant, which only depends on the nature of the sampling process, *i.e.*, how the dataset is collected and constructed. $\square$

### C.3 Proof of proposition 3

Assume that the feature representation $\hat{X}$ has zero mean. Then, $I(X; \hat{X}|Y) = I(X; \sigma(\hat{X})|Y)$.

*Proof.* The softmax function for an input vector $x \in \mathbb{R}^N$ is defined as

$$\sigma(x)[i] = \frac{e^{x[i]}}{\sum_{j \in [N]} e^{x[j]}}. \tag{51}$$

Following the proof of proposition 1 in section C.1, we aim to show that the softmax function is injective if its domain is in the subspace with zero mean. Assume two vectors $x; y \in \mathbb{R}^N$, such that

$$\sum_{i \in [N]} x_i = 0, \quad \sum_{i \in [N]} y_i = 0, \tag{52}$$

$$\frac{e^{x[i]}}{\sum_{j \in [N]} e^{x[j]}} = \frac{e^{y[i]}}{\sum_{j \in [N]} e^{y[j]}}. \tag{53}$$

Rewriting eq. (53),

$$e^{x[i]} \sum_{j \in [N]} e^{y[j]} = e^{y[i]} \sum_{j \in [N]} e^{x[j]}, \quad \forall i \in [N]. \tag{54}$$

We define the ratio of $\sum_{j \in [N]} e^{x[j]}$ and $\sum_{j \in [N]} e^{y[j]}$ as:

$$\Theta = \frac{\sum_{j \in [N]} e^{x[j]}}{\sum_{j \in [N]} e^{y[j]}}. \tag{55}$$

Substitute the eq. (55) into eq. (54) and take logarithm on both sides:

$$x_i = \log \Theta + y_i. \tag{56}$$

Since both $x$ and $y$ have zero mean, we have:

$$\sum_{i \in [N]} x_i = \sum_{i \in [N]} \log \Theta + y_i = 0, \tag{57}$$

$$N \log \Theta = 0, \ \Theta = 1. \tag{58}$$

Thus $x_i = y_i, \ \forall i \in [N]$. □

### C.4 Proof of proposition 4

For a encoder $f$ parametrized by $\theta$

$$\min_{\theta} H(X|\hat{X}, Y) \equiv \max_{\theta} I(X; \hat{X}).$$

*Proof.*

$$\text{I}(X; \hat{X}) = \text{H}(X) - \text{H}(X|\hat{X}) \tag{59}$$
$$= \text{H}(X) - \text{H}(X|\hat{X}, Y) \tag{60}$$

where $\text{H}(X)$ is a constant, which equals the amount of information in the dataset, eq. (60) is due to $Y \to X \to \hat{X}$ forms a Markov chain. $\qquad\square$

### C.5 Proof of proposition 5

*Proof.* Consider a random variable $Y$, with $N$ classes. Its entropy is given by

$$\text{H}(Y) = - \sum_{n \in [N]} P_n \log P_n. \tag{61}$$

Without losing generality, suppose we split the first sample point $y_1$ into two sample points $\hat{y}_a$ and $\hat{y}_b$, such that

$$P[y_1] = P[\hat{y}_a] + P[\hat{y}_b], \ s.t. \ P[\hat{y}_a] > 0; P[\hat{y}_b] > 0, \tag{62}$$

This transformation produces a new random variable $\hat{Y}$ with $N + 1$ sample points. The change in entropy is then

$$\text{H}(\hat{Y}) - \text{H}(Y) \tag{63}$$
$$= - P[\hat{y}_b] \log P[\hat{y}_b] - P[\hat{y}_a] \log P[\hat{y}_a] + P[y_1] \log P[y_1] \tag{64}$$
$$= - P[\hat{y}_b] \log P[\hat{y}_b] - P[\hat{y}_a] \log P[\hat{y}_a] + \{P[\hat{y}_b] + P[\hat{y}_a]\} \log\{P[\hat{y}_b] + P[\hat{y}_a]\} \tag{65}$$
$$= - P[\hat{y}_b]\big\{ \log P[\hat{y}_b] - \log\{P[\hat{y}_b] + P[\hat{y}_a]\}\big\} - P[\hat{y}_a]\big\{ \log P[\hat{y}_a] - \log\{P[\hat{y}_b] + P[\hat{y}_a]\}\big\} \tag{66}$$
$$> 0. \tag{67}$$

Thus $\text{H}(\hat{Y}) > \text{H}(Y)$, that is, splitting a single sample point into two distinct points increases the entropy of a random variable. In other words, given a set of samples, the entropy $\text{H}(Y)$ is maximized when each sample is assigned to a unique class. Conversely, for a fixed number of sample points, the entropy is maximized when their probabilities are uniformly distributed. Therefore, the entropy $\text{H}(Y)$ is maximized if the problem is formulated as instance discrimination. $\qquad\square$

## D  Further discussion on VE and information bottleneck

In this section, we discuss the relationship between the maximized mutual information method and information bottleneck, and provide an alternative approach to derive the objective function for the VE method.

### D.1 Relationship Between VE and Information Bottleneck

We depict the Venn diagram of which show the relationships between the $\text{H}(X)$, prototype information $\text{I}(X;Y)$, contextual information $\text{H}(X|Y)$ and $\text{I}(\hat{X};X)$ by the encoder trained by information bottleneck (a) and VE (b) in fig. 7.

Information bottleneck aims to minimize the following objective function:

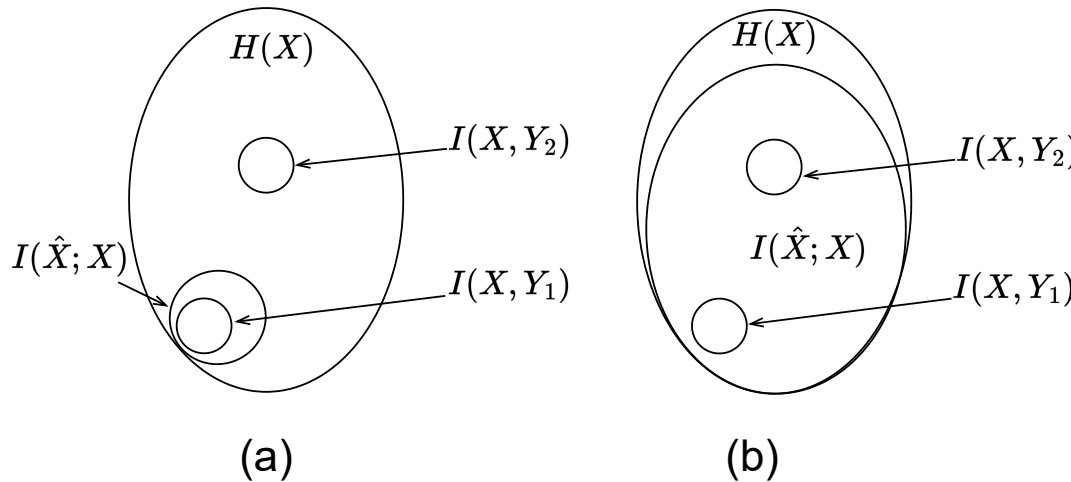

Figure 7: Mutual information between $\hat{X}$ and $X$ for models trained with objectives of (a) information bottleneck and (b) VE (ours).

$$\min \mathrm{I}(X; \hat{X}) - \beta \mathrm{I}(\hat{X}; Y), \tag{68}$$

which can be interoperated as finding a compressed representation $\hat{X}$ of $X$ that retains as much information about $Y$ as possible, while minimizing the information retained from $X$.

While the target of VE differs from the information bottleneck, as it aims to maximize the mutual information $\mathrm{I}(X; \hat{X})$, as though, the compressed representation $\hat{X}$ retains as much information about $X$ as possible.

### D.2 Alternative Approach to Simplify the Prototype Information

With a slight abuse of notation, in this section, we refer to $\hat{Y}$ as the label predicted from the feature $\hat{X}$.

$$\mathrm{I}(\hat{X}; Y) = \mathrm{H}(Y) - \mathrm{H}(Y|\hat{X}) \tag{69}$$
$$\geq \mathrm{H}(Y) - \mathrm{H}(E) - P(E)\log(|Y| - 1) \tag{70}$$

$$\geq \mathrm{H}(Y) - \mathrm{H}(E) - \log(|Y| - 1)\mathbb{E}_X\left[1 - \sum_{i=1}^{|Y|} P_{Y|X}(i|x)P_{\hat{Y}|X}(i|x)\right] \tag{71}$$

$$= \mathrm{H}(Y) - \mathrm{H}(E) - \log(|Y| - 1)\mathbb{E}_X\left[\sum_{i=1}^{|Y|} P_{Y|X}(i|x)\Big[1 - P_{\hat{Y}|X}(i|x)\Big]\right] \tag{72}$$

$$\geq \mathrm{H}(Y) - \mathrm{H}(E) - \log(|Y| - 1)\mathbb{E}_X\left[\sum_{i=1}^{|Y|} -P_{Y|X}(i|x)\log P_{\hat{Y}|X}(i|x)\right] \tag{73}$$

$$= \mathrm{H}(Y) - \mathrm{H}(E) - \log(|Y| - 1)\mathbb{E}_X[\mathrm{H}(P_{Y|X}(i|x), P_{\hat{Y}|X}(i|x))], \tag{74}$$

where eq. (70) follows from (Hartley, 1928), $\log(|Y| - 1)$ is a constant, and $\mathrm{H}(E)$ approaches zero when no prediction error occurs.

## E  Running Time

In this section, we report the running time of the IGDS algorithm and compare its efficiency with the MiniMax (Gu et al., 2024a). To this end, we sampled 100 images with resolution $256 \times 256$ for the ImageWoof and ImageNette datasets using both methods on the clusters we used; all comparisons are done with one single NVIDIA V100 GPU. The running time is reported in the table 7.

Table 7: The running time of MiniMax Diffusion and IGDS.

| Dataset | ImageNette | ImageWoof |
|---------|------------|-----------|
| MiniMax | 44 mins | 46 mins |
| IGDS | 57 mins | 53 mins |

Compared to MiniMax diffusion, IGDS slightly increases the time complexity, primarily due to the additional VE model required in the distillation process.

## F  Evaluation Protocol

We report the evaluation protocol in this section. Three commonly used network architectures are used for evaluation:

• **ConvNet-6**, a 6-layer convolutional network, is an extension of ConvNet-3, which is commonly used in previous dataset distillation (DD) works for small-resolution images. To accommodate full-sized $256\times256$ ImageNet data, we add three additional layers. Each layer contains 128 feature channels, and instance normalization is applied.

• **ResNetAP-10** is a 10-layer ResNet variant in which the standard strided convolution is replaced with average pooling for downsampling, allowing for smoother feature aggregation.

• **ResNet-18** is an 18-layer ResNet modified to use instance normalization (IN) instead of batch normalization. Since IN performs better than batch normalization under our experimental protocol, we adopt it consistently across all ResNet-18 models.

During the evaluation training, we closely follow the protocols established in (Kim et al., 2022b; Gu et al., 2024a). Specifically, we use the Adam optimizer with a fixed learning rate of 0.01 across all experiments to ensure consistency in optimization. The number of training epochs for different IPC settings is detailed in Table 8. The applied data augmentations are random resize-crop, random horizontal flip, and CutMix (Yun et al., 2019).

Table 8: Evaluation training epochs across different IPC settings.

| IPC | 1 | 10 | 20 | 50 | 100 |
|-----|------|------|------|------|------|
| Epochs | 2000 | 2000 | 1500 | 1500 | 1000 |

## G  Transfer Learning Results

Real-world ML pipelines repeatedly pretrain and adapt models to new tasks under tight compute, data access, and architecture constraints. We therefore evaluate our method in a transfer-learning setting to verify that it learns task-agnostic representations that transfer well while reducing training cost. Following Lee et al. (2024), we pretrain on a distilled source set and fine-tune on labeled targets. Specifically, we distill CIFAR-100 to 1,000 images at $32 \times 32$ using a 3-layer ConvNet.

After distillation, we pretrain the ConvNet for 1,000 epochs with SGD (learning rate 0.1, momentum 0.9, weight decay $10^{-3}$, batch size 256). We then fine-tune on CIFAR-10/100, FGVC-Aircraft, Stanford Cars, CUB-200-2011, Stanford Dogs, and Flowers-102 for 10k iterations (5k for Aircraft, Cars, CUB, Dogs, and

Table 9: The results of transfer learning with CIFAR-100. The data compression ratio for source data is 2%. ConvNet3 is pre-trained on a condensed dataset and then fine-tuned on target datasets. We report the average and standard deviation over three runs. The best results are bolded.

| Method | Source CIFAR100 | Target CIFAR10 | Aircraft | Cars | CUB2011 | Dogs | Flowers |
|--------|-----------------|----------------|----------|------|---------|------|---------|
| Random | $65.23_{\pm 0.21}$ | $87.55_{\pm 0.19}$ | $33.99_{\pm 0.45}$ | $19.77_{\pm 0.21}$ | $18.18_{\pm 0.21}$ | $21.69_{\pm 0.18}$ | $59.31_{\pm 0.27}$ |
| MTT | $65.92_{\pm 0.18}$ | $87.87_{\pm 0.08}$ | $36.11_{\pm 0.27}$ | $21.42_{\pm 0.03}$ | $18.94_{\pm 0.41}$ | $22.82_{\pm 0.02}$ | $60.88_{\pm 0.45}$ |
| KRR-ST | $66.81_{\pm 0.11}$ | $88.72_{\pm 0.11}$ | $\mathbf{41.54_{\pm 0.37}}$ | $28.68_{\pm 0.32}$ | $\mathbf{25.30_{\pm 0.37}}$ | $26.39_{\pm 0.08}$ | $67.88_{\pm 0.18}$ |
| IGDS | $\mathbf{67.02_{\pm 0.07}}$ | $\mathbf{91.72_{\pm 0.31}}$ | $40.86_{\pm 0.21}$ | $\mathbf{28.73_{\pm 0.24}}$ | $25.21 \pm 0.53$ | $\mathbf{27.51 \pm 0.18}$ | $\mathbf{68.02 \pm 0.20}$ |

Flowers) using SGD (learning rate 0.01, momentum 0.9, weight decay $5 \times 10^{-4}$) with cosine learning-rate decay and batch size 256. We report Top-1 accuracy averaged over three random seeds.

As shown in Table 9, we benchmark IGDS against two dataset-distillation baselines (MTT (Cazenavette et al., 2022b) and KRR-ST (Lee et al., 2024)) and a randomly selected subset. Although IGDS was not designed specifically for transfer learning, it matches or exceeds KRR-ST on most targets and achieves the best average performance across datasets. This suggests that IGDS distills task-agnostic features that transfer well and supports our hypothesis that controlling contextual information during distillation improves downstream generalization.

## H    Samples Generated by IGDS

In this section, we provide additional examples of distilled datasets for ImageNette and ImageWoof with IPC 10, shown in fig. 8 and fig. 9, respectively.

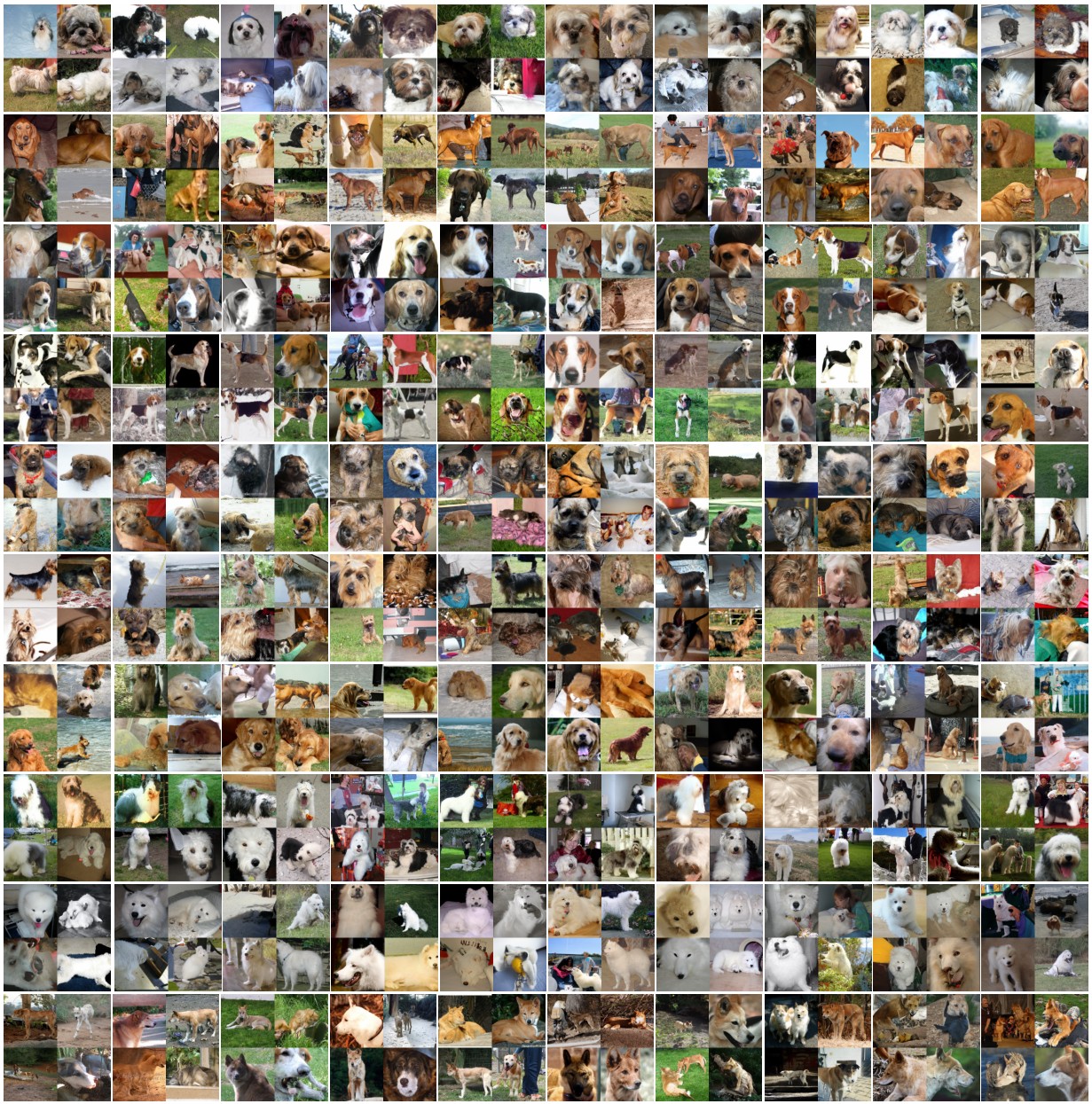

Figure 8: Distilled Image Visualization: ImageNette with IPC 10.

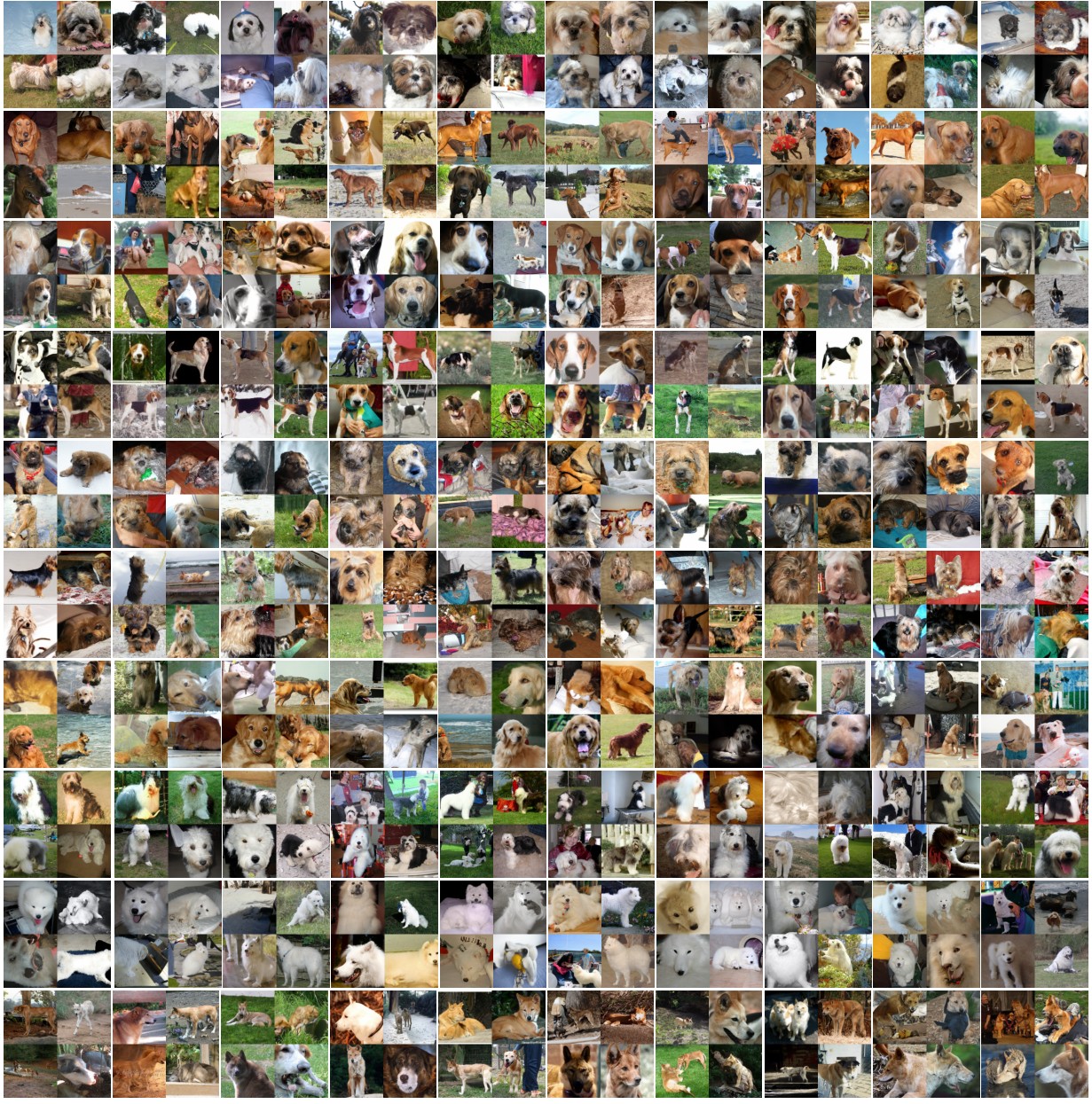

Figure 9: Distilled Image Visualization: ImageWoof with IPC 10.

