# OpenReview forum: "Information-Guided Diffusion Sampling for Image Classification Dataset Distillation"
_TMLR — Rejected by TMLR_

### Review · Reviewer_7Cq1 · 2025-08-08

**Summary Of Contributions:**

The authors propose a method for dataset distillation by generating synthetic datapoints using a diffusion model trained with an information-theoretic loss. Specifically, the loss balances the mutual information between inputs and labels with the conditional entropy of each class. These quantities are estimated using variational approximations.

**Strengths**

* Dataset distillation is a timely and relevant problem, with broad potential applications in efficient learning and data privacy.
* The use of information-theoretic principles offers an elegant and principled approach to the problem.
* The decision to release code is commendable and contributes to the reproducibility and accessibility of the work.

**Weaknesses**

1. *Overreliance on vision-specific framing:*
   The paper is unnecessarily anchored in the context of image classification and is heavily laden with computer vision jargon. For example, renaming standard terms such as "mutual information" and "conditional entropy" to "prototype information" and "contextual information" seems gratuitous and may obscure the theoretical contributions rather than clarify them. Similarly, some notation (e.g., using @ for matrix multiplication) and the pseudocode in Algorithm 1 is overly similar to regular code when, in my opinion, a more mathematical exposition would be easier to follow.

2. *Lack of clarity in problem setup:*
   The presentation of the distillation task lacks rigor, e.g. it is not clearly defined what the random variables are and, in particular, if the associated probability distributions are the empirical ones or the (approximate or ground truth) continuous ones. This ambiguity hampers the reader’s ability to interpret the proposed objective and its implications. In particular, it is difficult to understand why $\beta=1$ would not be the default choice if the goal is to retain all information from the original dataset.

3. *Scope and generality are not clearly articulated:*
   While the method is evaluated only on classification tasks, it is not clear whether the approach is inherently limited to classification. Moreover, if the task is known, it is unclear why preserving the conditional entropy (contextual information) is desirable. If the intention is to restrict the scope to image classification, I recommend adapting the title and abstract accordingly to avoid overclaiming.

4. *Insufficient connection to prior work on information bottleneck and variational MI estimation:*
   Although the paper briefly mentions the connection to the information bottleneck principle, the discussion is too cursory to position the work meaningfully within that literature and to applications such as neural compression. Likewise, there is limited comparison to existing variational methods for mutual information and conditional entropy estimation, such as "Deep variational information bottleneck" (Alemi et al., 2017) and "Mutual Information Neural Estimation" (Belghazi et al., 2018). To support the authors' claims, I'd like to see both a theoretical explanation of the differences and experiments that compare the accuracy of the estimates in isolation from their use in the overall framework.

5. *Experimental section lacks structure and depth:*
   The experiments are fragmented and do not clearly convey what specific questions are being addressed. The paper would benefit significantly from a clearer experimental design and narrative. Ideally, the section should be organized into subsections that address distinct research questions—for example:

* How accurate are the variational estimates of mutual information and conditional entropy?
* What is the effect of varying beta?
* How does the distilled dataset compare to baselines in downstream task performance?

**Audience:**

Yes

**Audience Explanation:**

See strengths above.

**Claims And Evidence:**

No

**Claims Explanation:**

See weaknesses above.

**Requested Changes:**

To strengthen the paper and improve its clarity and impact, I suggest the following:

1. Clarify the probabilistic setup. Define all random variables, their associated distributions, and whether these are empirical or continuous. Make explicit the assumptions about access to labels or class distributions during distillation.

2. Use standard terminology. Avoid introducing non-standard terms ("prototype information", etc.) unless they offer genuine conceptual clarity. Stick to established information-theoretic terminology to better align with prior work. Reduce the amount of abbreviations.

3. Position the method more clearly. If the method is limited to classification, revise the title and abstract accordingly. Otherwise, clarify how the framework might extend to other tasks and justify the role of conditional entropy in the objective.

4. Expand discussion of related work. Include a more thorough comparison to relevant literature on variational estimation of mutual information and the information bottleneck, both conceptually and empirically.

5. Improve experimental design. Reorganize the experimental section to follow a set of clearly posed questions or hypotheses. Include ablation studies and quantitative evaluation of the information estimates in isolation.

6. Revise notation and presentation. Use mathematically standard notation throughout, and avoid implementation-specific symbols unless absolutely necessary.

---

> ### Author Response · Authors · 2025-10-14
>
> Thank you for your detailed review comments. We agree that dataset distillation is a timely and relevant problem and, to us, an essential lens for understanding the learning process. To the best of our knowledge, we are the first to study the dataset distillation through the lens of classical information theory, which as you noted, yields an elegant variational estimator for two information theory quantities: conditional mutual information and conditional entropy. Below, we address your concerns.
>
> **W1a. Overreliance on vision-specific framing: The paper is unnecessarily anchored in the context of image classification**
>
> Thank you for pointing this out. We would like to clarify that, in most DD papers, the default task is image classification; authors often do not explicitly state the task in the title when targeting image-classification DD. We acknowledge that the DD field  is moving quickly as more modalities and tasks are included. Positioning the paper has therefore become increasingly important, and clearly defining its scope is essential for situating it in the literature.  Accordingly, we changed the title of the paper to “Information-Guided Diffusion Sampling for Image Classification Dataset Distillation.”
>
> **W1b.The paper is heavily laden with computer vision jargon.**
>
> Thank you for the feedback. We clarify that “prototype information” and “contextual information” refer to mutual information and conditional entropy, respectively,  as these terms capture the idea that image-classification datasets contain two distinct forms of information, and have been accepted by both machine learning and information theory communities [1]. To aid understanding, we visualize examples with both high and low contextual information in Section 5.3, “Semantic Meaning of Prototype and Contextual Information”. Additionally, we chose to use “prototype information” and “contextual information” instead of the more formal expressions—“mutual information between the ground truth label and the input” and “conditional entropy of the input given its ground truth label”—for the sake of brevity. We revise the footnote to make it clear.
> [1] Bayes conditional distribution estimation for knowledge distillation based on conditional mutual information; Linfeng Ye, Shayan Mohajer Hamidi, Renhao Tan, En-Hui Yang, 2024 International Conference on Learning Representations2024 International Conference on Learning Representations.
>
> **W1c.  Similarly, some notation (e.g., using @ for matrix multiplication) and the pseudocode in Algorithm 1 is overly similar to regular code when, in my opinion, a more mathematical exposition would be easier to follow.**
>
> Thank you for pointing this out. We have updated the matrix multiplication notation in the pseudocode and notation section accordingly.
>
> **W2. Lack of clarity in problem setup: The presentation of the distillation task lacks rigor.**
>
> Thank you for pointing this out. The random variables in our work are well defined. See Section 3 Notation. The associated distributions are continuous —reflecting the nature of images— and can be approximated using the dataset, which is a standard practice in machine learning research.
>
> **W3a. Scope and generality are not clearly articulated: While the method is evaluated only on classification tasks, it is not clear whether the approach is inherently limited to classification. If the intention is to restrict the scope to image classification, I recommend adapting the title and abstract accordingly to avoid overclaiming.**
>
> Yes, our work builds on the Conditional Mutual Information (CMI) framework [1]. Extending it to other tasks (e.g., regression) requires revising the theory. We have revised the title and the abstract of the paper to avoid overclaiming.
> [1] Conditional Mutual Information Constrained Deep Learning for Classification, En-Hui Yang; Shayan Mohajer Hamidi; Linfeng Ye; Renhao Tan; Beverly Yang, IEEE Transactions on Neural Networks and Learning Systems, 2025,Volume: 36 Issue: 8.
>
> **W3b. Moreover, if the task is known, it is unclear why preserving the conditional entropy (contextual information) is desirable.**
>
> We preserve contextual information (i.e., conditional entropy) because the original dataset contains two complementary types of information: prototype and contextual. To construct an informative distilled dataset, we must include both. Moreover, we observe that the optimal amount of contextual information varies across different IPC settings, which further suggests the need to control the contextual information accordingly under different IPC settings.

---

> > ### Author Response · Authors · 2025-10-14
> >
> > **W4. Insufficient connection to prior work on information bottleneck and variational MI estimation: Although the paper briefly mentions the connection to the information bottleneck principle, the discussion is too cursory to position the work meaningfully within that literature and to applications such as neural compression.**
> >
> > Thank you for your suggestions. As we discussed in the main text and appendix, our framework fundamentally differs from the Information Bottleneck (IB) principle. Specifically, our approach seeks to preserve as much dataset information as possible before compression by the classifier, enabling the variational estimator to provide more precise guidance for image generation. By contrast, IB explicitly compresses the input so that the intermediate representation retains only class-relevant information. We have cited both papers in the discussion and expanded the related-work section to cover related topics like neural compression.
> >
> > **W4b. I'd like to see both a theoretical explanation of the differences and experiments that compare the accuracy of the estimates in isolation from their use in the overall framework.**
> >
> > To the best of our knowledge, we are the first to (i) decompose dataset information into prototype and contextual information; (ii) develop a novel variational estimator for both; and (iii) apply the resulting variational lower bound to dataset distillation. Existing variational methods for mutual information and conditional entropy primarily target the information bottleneck or a network’s mutual information, rather than providing tight lower bounds for prototype and contextual information in the dataset; as a result, they cannot be directly used to estimate both information. If the reviewer has specific experimental suggestions for applying those methods in our setting, we would be happy to test them.
> >
> > **W5a. Experimental section lacks structure and depth: The experiments are fragmented and do not clearly convey what specific questions are being addressed.**
> >
> > To improve clarity, we have reorganized the Experimental section into the following subsections:
> >
> > 5. Experiments
> >
> > 	5.1 Experimental setup
> >
> > 	5.2  How to select β in eq. (25)
> >
> > 	5.3 Semantic Meaning of Prototype and Contextual Information
> >
> > 	5.4 Comparison with State-of-the-art Methods
> >
> > 	5.5 Cross-architecture performance
> >
> > 	5.6 Combining with Priors Beyond DDPM
> >
> > 	5.7 Tightness of the Lower Bounds
> >
> > We also include experiments evaluating the performance on downstream tasks in the appendix.
> >
> > **W5b. How accurate are the variational estimates of mutual information and conditional entropy?**
> >
> > We address this research question as follows:
> > Since directly measuring the value of I(X;Y) and H(X|Y) is infeasible for a natural-image dataset, and it is impossible to certify the tightness of these quantities on real data because the true data-generating distribution is unknown. This limitation motivates our novel training procedure for learning a variational estimator. To verify that our method indeed tightens the estimates of the I(X;Y) and H(X|Y), we conduct a set of toy experiments and report the results in Section 5.7. Specifically, we generate three 2-dimensional Gaussian distributions in space, train an encoder on these samples using the proposed method, then measure the I(X;Y) and H(X;Y), and compare against an identical model trained with cross-entropy for classification. In summary, our variational training approach yields tighter lower bounds on both quantities than the cross-entropy–trained baseline.
> >
> > The results are presented in section 5.7.
> >
> > **W5c. How does the distilled dataset compare to baselines in downstream task performance?**
> >
> > We acknowledge that downstream performance is an important criterion for evaluating a distilled dataset. Accordingly, we assess IGDS under a transfer-learning setting; results are provided in Appendix G. Although IGDS was not designed specifically for transfer learning, it matches or surpasses prior state-of-the-art on most transfer datasets and achieves the best average performance across datasets. These results suggest that IGDS distills task-agnostic features that transfer well, supporting our observation that controlling contextual information during distillation improves dataset quality.

---

> > > ### Author Response · Authors · 2025-10-14
> > >
> > > **Requested Changes:**
> > >
> > > **RC1. Clarify the probabilistic setup. Define all random variables, their associated distributions, and whether these are empirical or continuous. Make explicit the assumptions about access to labels or class distributions during distillation.**
> > >
> > > We have revised the Notation and Preliminaries section to clarify assumptions about access to labels and class distributions during distillation.
> > >
> > > **RC2.  Use standard terminology. Avoid introducing non-standard terms ("prototype information", etc.) unless they offer genuine conceptual clarity. Stick to established information-theoretic terminology to better align with prior work. Reduce the amount of abbreviations.**
> > >
> > >  Please refer to our response to Weakness W1b.
> > >
> > > **RC3. Position the method more clearly.**
> > >
> > > We have revised the abstract and changed the title to “Information-Guided Diffusion Sampling for Dataset Distillation in Image Classification” and revised the abstract accordingly.
> > >
> > >
> > > **RC4a. Expand discussion of related work.**
> > >
> > >  We have expanded the Related Work section to include research on neural compression, and we discuss extensions to other tasks in the Future Work section.
> > >
> > > **RC4b. Include a more thorough comparison to relevant literature on variational estimation of mutual information and the information bottleneck, both conceptually and empirically.**
> > >
> > > Please refer to our response to Weakness W4b.
> > >
> > > **RC5a. Improve experimental design. Reorganize the experimental section to follow a set of clearly posed questions or hypotheses.**
> > >
> > > We have revised the experimental section accordingly. All the hypotheses have been clearly declared in the submission.
> > >
> > > **RC5b. Include ablation studies.**
> > >
> > > An ablation study is provided in Section 6 of the original submission. To further elucidate the semantic roles of contextual and prototype information, we analyze how β influences the distilled images (Figures 3–4).
> > >
> > > **RC5c.  Include  quantitative evaluation of the information estimates in isolation.**
> > >
> > > In general, it is challenging to estimate the tightness of the lower bound on the real-world datasets, as the original dataset distribution is generally unavailable, which is the motivation for proposing the variational estimator to estimate both information quantities.
> > > We empirically quantify tightness on a synthetic Gaussian dataset where ground-truth I(X;Y) and H(X|Y) are available.  As reported in Sec. 5.7, our variational estimator yields substantially tighter bounds than a cross-entropy-trained baseline across all settings. Supporting the effectiveness of our approach.
> > >
> > > **RC6. Revise notation and presentation. Use mathematically standard notation throughout, and avoid implementation-specific symbols unless absolutely necessary.**
> > >
> > > We have changed the notation of matrix multiplication to $AB$.

---

### Review · Reviewer_GygP · 2025-08-23

**Summary Of Contributions:**

The authors claim to introduce a dataset distillation method where diffusion models are used to sample the distilled dataset. The authors suggest using a guidance term based in information theory to ensure that the distilled dataset is diverse and informative.

Overall, this is a very poorly written paper. Most of the paper is spent on derivations of an estimator for information-theoretical quantities without a clear explanation of why each derivation is being carried out. The main point of the paper is to use the derived bounds for guidance in diffusion models, yet diffusion models are not covered in the background section, and we do not even get an equation specifying in a precise manner the guidance term used in the sampling process. The paper also contains what I believe are several errors and formatting issues.

**Audience:**

No

**Audience Explanation:**

Although the topic of data distillation is of interest to some members of TMLR's community, in its current state, this paper is unclear enough to not be of interest to TMLR's audience.

**Broader Impact Concerns:**

I have no broader impact concerns.

**Claims And Evidence:**

No

**Claims Explanation:**

Given the lack of clarity and the poor writing in the paper, it is hard to even say, precisely, what the main claims being made are. As such, the authors do not provide enough evidence to support their claims.

To highlight but a few issues affecting clarity:

- The notation $g_\psi:\{ \hat{X} \rightarrow \hat{Y} | \hat{Y} = \psi \hat{X} \}$ doesn't mean anything; I assume the authors meant $g_\psi(\hat{X}) = \hat{Y} = \psi \hat{X}$.

- It is claimed after proposition 2 that you train $f$ and $g$ to maximize $I(X, \hat{X})$, yet this mutual information is affected only by $f$, not by $g$. This also does not seem to be a typo since the claim is repeated throughout the paper.

- Section 3 introduces notation which is never used, yet does not introduce notation which you actually do use.

- Diffusion models are not covered in the background, nor is it made precise how anything in the paper is used along with diffusion models.

- "where $\tilde{Y}$ is an auxiliary variable denoting the task type": this sentence is impossible to understand without having defined what the task type is.

**Requested Changes:**

I do not have any requested changes. I believe that the changes this paper needs before it can be published are too large to address in a rebuttal, and that it should be completely rewritten.

---

> ### Author Response · Authors · 2025-10-14
>
> Thank you for your constructive comments. Our point-by-point reply is as follows.
>
> **Overall, this is a very poorly written paper.**
>
> We respectfully disagree with the assertion that the paper is poorly written.  Reviewer snbf notes that the paper is easy to follow and well written, and Reviewer 7Cq1 indicates a clear understanding of most of our contributions. To help convey the core ideas, we summarize our main contributions below.
>
> ***Theoretical contributions:***
>
> (a) To the best of our knowledge, we are the first to decompose the dataset information into contextual information (conditional entropy H(X|Y)) and prototype information (mutual information I(X;Y)).
>
> (b) We provide the first method to train a variational estimator to jointly estimate both H(X|Y) and I(X;Y)).
>
> (c) We are the first to identify the proportional relationship between the optimal contextual information and IPC–a result that is valuable for the dataset distillation community and has broader implications for areas such as dataset curation.
>
> ***Experimental contributions:***
>
> IGDS achieves state-of-the-art performance across multiple datasets. We also observe that the optimal trade-off between contextual and prototype information varies with the IPC setting.
>
> To the best of our knowledge, we are the first to propose variational estimators for estimating the contextual and prototype information of a given dataset, and we further observe that the optimal contextual information is directly proportional to the IPC of distilled datasets.
>
> The paper’s contributions are explicitly itemized at the end of the Introduction as bullet points.
>
> In light of these points, we respectfully invite the reviewer to reconsider the assessment of our submission.
>
> **Most of the paper is spent on derivations of an estimator for information-theoretical quantities without a clear explanation of why each derivation is being carried out.**
>
> The derivations in this paper serve three purposes, to specify the exact information quantities we estimate, to prove they are computable lower bounds with identifiable slack terms, and to show precisely how each quantity is used in the training or guidance objective.
>
> **The main point of the paper is to use the derived bounds for guidance in diffusion models**
>
> We note that the bound itself cannot guide the diffusion model. Instead, we rely on a pretrained encoder—trained with a novel variational method for estimating both contextual and prototype information—to provide this guidance.
>
> **Diffusion models are not covered in the background section.**
>
> We respectfully disagree with the assertion that diffusion models are not covered in the background section. We discuss the diffusion model in the  Related Work section in our original submission.
>
> **We do not even get an equation specifying in a precise manner the guidance term used in the sampling process.**
>
> Algorithm 2 details IGDS sampling. At line 13, the diffusion process is guided by the gradient of the IGDS loss; this is now explicitly noted in the manuscript.
>
> **The paper also contains what I believe are several errors and formatting issues.**
>
> We appreciate the reviewer’s attention to typographical issues and have corrected them in the revised version. However, formatting issues were not identified among the stated weaknesses.
>
> **Given the lack of clarity and the poor writing in the paper, it is hard to even say, precisely, what the main claims being made are.**
>
> Thank you for the feedback. In the original submission, we explicitly state the main claims and list our contributions at the end of the Introduction. We recognize that parts of the manuscript could be clearer. In the revision, we have edited the paper for clarity throughout. We hope these changes address your concern.
>
> **The notation $g_\psi: \hat{X}\rightarrow \hat{Y}|\hat{Y}=\psi \hat{X}$ doesn’t mean anything; I assume the authors meant $g_\psi(\hat{X}) = \hat{Y} = \psi \hat{X}$.**
>
> Thank you for pointing it out. We are following the set-theoretic definition of a function. The notation has been revised for better readability.
>
> **It is claimed after proposition 2 that you train $f$ and  $g$ to maximize $I(X,\hat{X})$, yet this mutual information is affected only by $f$, not by $g$. This also does not seem to be a typo since the claim is repeated throughout the paper.**
>
>
> We train $f$ to maximize the $I(X,\hat{X})$, where $\hat{X}$ is the random variable representing the output of $f$. This objective encourages $\hat{X}$ to encode as much information about the input $X$ as possible, thus $f$ can estimate contextual information.  In contrast, g is designed to discard non-prototype information, yielding a more precise estimate of prototype information; accordingly, using $\hat{Y}$ to estimate contextual information is suboptimal.

---

> > ### Author Response · Authors · 2025-10-14
> >
> > **Section 3 introduces notation which is never used, yet does not introduce notation which you actually do use.**
> >
> > Thank you for pointing it out. After carefully proofreading the paper, we found that the only unused notation was the cross-entropy, which has been removed in the revised version.
> >
> > **Diffusion models are not covered in the background, nor is it made precise how anything in the paper is used along with diffusion models.**
> >
> > We respectfully disagree with your statement: in the original submission, we discussed the diffusion model in two subsections. Specifically, Section 2.2 explains how to perform dataset distillation using generative models, and Section 2.3 explicitly discusses the diffusion model for completeness.
> >
> > **"where is an auxiliary variable denoting the task type": this sentence is impossible to understand without having defined what the task type is.**
> >
> >  Thank you for pointing it out. We have revised the text for clarity. \tilde{Y} is an auxiliary random variable whose distribution we are free to specify; in our setting, it denotes the class assignment in the classification task.   In particular,  when we seek to maximize the H(\tilde{Y}), we assign a unique label to each training sample in the dataset (see Proposition 5), which leads to the instance discrimination task.

---

> > > ### Comment · Reviewer_GygP · 2025-10-30
> > >
> > > Thank you for your attempts at clarifying my concerns. Unfortunately, all of my concerns remain: the contributions and notation are unclear, the bulk of the paper is spent on derivations which are not properly motivated to the reader, and I am still unsure of the correctness of the paper due to its lack of clarity.
> > >
> > > As for $I(X, \hat{X})$, I understand $\hat{X}$ is the output of $f$. What I mean is that neither this output nor $X$ depend on $g$, so I do not see how this mutual information can be maximized over both $f$ and $g$.
> > >
> > > You also claim that you did discuss diffusion models. I know you wrote a paragraph about them, but this paragraph has no equations whatsoever and no clear explanation of how you apply guidance.

---

> ### Author Response · Authors · 2025-11-02
>
> ## Unfortunately, all of my concerns remain: the contributions are unclear
>
> To help the reviewer understand the contribution and significance of the work, we first elaborate on the motivation behind our research. As discussed in the introduction, we aim to study the role of different information in the distilled dataset. To this end, we first decompose the information into conditional entropy (prototype information) and conditional mutual information (contextual information), and then, how does each of these types of information contribute to the distilled dataset? We empirically found that the optimal contextual information varies across the IPC, specifically that under large image per class (IPC) settings, preferences for more contextual information are observed, while in small IPC settings, preferences for fewer contextual information are observed. Estimating the contextual information and prototype information is non-trivial. To this end, we derive two variational lower bounds, which, for the sake of clarity, occupy most of the paper. To help the reviewer grab the main idea, we have listed all the contributions in our previous discussion.
> ​In order to study the role of different information in the distilled dataset. We have the following contribution, which has been listed in the previous reply:
>
> Theoretical contributions:
>
> (a) To the best of our knowledge, we are the first to decompose the dataset information into contextual information (conditional entropy H(X|Y)) and prototype information (mutual information I(X;Y)).
>
> (b) We provide the first method to train a variational estimator to jointly estimate both H(X|Y) and I(X;Y)).
>
> (c) We are the first to identify the proportional relationship between the optimal contextual information and IPC–a result that is valuable for the dataset distillation community and has broader implications for areas such as dataset curation.
> Experimental contributions:
>
> IGDS achieves state-of-the-art performance across multiple datasets. We also observe that the optimal trade-off between contextual and prototype information varies with the IPC setting.
> To the best of our knowledge, we are the first to propose variational estimators for estimating the contextual and prototype information of a given dataset, and we further observe that the optimal contextual information is directly proportional to the IPC of distilled datasets.
>
> The paper’s contributions are explicitly itemized at the end of the Introduction as bullet points.
>
>
> ## notations are unclear
>
> We have proofread the paper and deleted the unused notation in the notation section. Could the reviewer please indicate which notation is still unclear?
>
> ## the bulk of the paper is spent on derivations which are not properly motivated to the reader
>
> We strongly disagree on the statement, before each derivation we have discussed the reason why we need it, for example, before the proposition 1, we clearly wrote that “The first term on the right-hand side of eq. (3), namely I(Xˆ; Y ), is difficult to compute directly. To overcome this difficulty, we introduce the following Proposition:”.
>
> ## As for $I(X, \hat{X})$, I understand $\hat{X}$ is the output of $f$. What I mean is that neither this output nor $X$ depend on $g$, so I do not see how this mutual information can be maximized over both $f$ and $g$.
>
> As we discussed in the previous reply, $I(X, \hat{X})$ is not dependent on $ g$, and we do not apply $I(X, \hat{X})$ to $ g$ either. The role of g is to assist in the estimation of the prototype information, as shown in Proposition 1. If the linear classifier has full column rank, then it can preserve the prototype information.
>
> ## You also claim that you did discuss diffusion models. I know you wrote a paragraph about them, but this paragraph has no equations
>
> Could the reviewer specify the request specifically?   We have clarified your previous concern, “Diffusion models are not covered in the background, nor is it made precise how anything in the paper is used along with diffusion models,” in the reply. In your previous comment, you did not request the inclusion of extra equations. In fact, diffusion is not the core contribution of our submission; this is the main reason why we devote only a small section to discussing it, for the sake of completeness of the paper.
> Could the reviewer please specify which equation you would like us to include?
>
>
> ##  no clear explanation of how you apply guidance.
>
> The method of applying guidance is clearly discussed in Section 4.2, and the pseudo-code effectively illustrates how to apply the variational estimator to guide the diffusion process. In the latest version, we make it explicit in Section 4.2.
>
> Given these points, we kindly request a reassessment of our manuscript.

---

> ### Comment · Reviewer_GygP · 2025-11-05
>
> Thank you again for the reply.
>
> - You asked me to clarify what my issues with the notation are: Proposition 3 uses the notation $\sigma(\hat{X})$ without ever introducing it. Shortly after, you use $Q^y$, and it is unclear if the first line in page 7 is defining $Q^y$ or providing an equality. Furthermore, the update you did to the notation in proposition 1 is still unclear: as written, you are treating $\hat{Y}$ as a set, which you do not in the rest of the paper. Less importantly, you define entropy but not mutual information.
>
> - You claim in the paper "requires training $f_\theta(\cdot)$ and $g_\psi(\cdot)$ to maximize $I(X; \hat{X})$"; this is either an extremely confusing typo (if I recall correctly, the first version of the paper made such a claim twice), or in contradiction with your previous reply.
>
> - Thank you for writing the guidance term more explicitly; this is (modulo my following concern) what I was asking for. Unfortunately I also found this rather confusing. The guidance term in eq 28 is a distribution-level quantity (i.e. it does not depend on a single sample), and I don't see how you can take its gradient wrt the sample being generated (eq 29). Overall this is confusing since one needs many samples to estimate mutual information, whereas guidance is applied to a single sample at a time. Clarifying this would also be helpful.

---

> > ### Author Response · Authors · 2025-11-09
> >
> > **You asked me to clarify what my issues with the notation are: Proposition 3 uses the notation \sigma{\hat{X}} without ever introducing it.**
> > We respectfully disagree. The Notation section has defined the softmax as $\sigma(\cdot)$;   we now restate this when it appears in Proposition 3 in the latest revision.
> >
> > **Shortly after, you use Q^y , and it is unclear if the first line in page 7 is defining Q^y  or providing an equality.**
> >
> > Thank you for your question. To clarify, the expression on page 7 is intended as a definition of Q^y, not an additional equality. Specifically, following [1], we define $Q^y=\frac{1}{|D_y|}\sum_{x\in D_y} \sigma(\hat{X})$. Accordingly, we have revised the text to ensure clarity.
> >
> > [1] En-Hui Yang, Shayan Mohajer Hamidi, Linfeng Ye, Renhao Tan, and Beverly Yang. Conditional mutual information constrained deep learning: Framework and preliminary results. In Proc. ISIT, pp. 569–574, 2024.
> >
> > **Furthermore, the update you did to the notation in proposition 1 is still unclear: as written, you are treating $\hat{Y}$ as a set, which you do not in the rest of the paper.**
> >
> > Thank you for pointing this out. It was a notational choice, and we have updated it per your previous suggestion.
> >
> > **Less importantly, you define entropy but not mutual information.**
> >
> > Thank you for your advice. The latest revision includes the definition of mutual information in the Notation section.
> >
> > **You claim in the paper "requires training $f$ and $g$ to maximize $I(X,\hat{X})$"; this is either an extremely confusing typo (if I recall correctly, the first version of the paper made such a claim twice), or in contradiction with your previous reply.**
> >
> > Thank you for the careful review. It improves the rigor of our work. The phrase "requires training $f$  and $g$ to maximize $I(X,\hat{X})$" was an unintended typo in the early draft.  Our method maximizes $I(X,\hat{X})$ with respect to $f$; $g$ is not trained in this step. We have verified that the current version no longer contains this error.
> >
> > **Thank you for writing the guidance term more explicitly; this is (modulo my following concern) what I was asking for. Unfortunately I also found this rather confusing. The guidance term in eq 28 is a distribution-level quantity (i.e. it does not depend on a single sample), and I don't see how you can take its gradient wrt the sample being generated (eq 29). Overall this is confusing since one needs many samples to estimate mutual information, whereas guidance is applied to a single sample at a time. Clarifying this would also be helpful.**
> >
> > Thank you for pointing out the disconnect between the distribution-level objective and per-sample guidance. During sampling, we maximize its mini-batch estimator $\hat{\mathcal{L}}_{IGDS}$ in the new Eq. 29.
> >
> > We revised the submission to make it clearer (Eqs. 29-31).

---

> ### Comment · Reviewer_GygP · 2025-11-12
>
> Thank you for your continued engagement.
>
> You are correct that $\sigma$ was defined, I apologize for having missed this. Overall, I still believe the paper is quite unclear and that it needs a major revision. I expand on some of the parts of the paper I still find unclear:
>
> - $H(Y|\hat{Y},Y)$ in eq 6 should be equal to 0, and I don't understand the point of this equation.
> - The expectation in equation 7 should also be wrt $\hat{Y}$ to be equal to eq 5.
> - You claim shortly after eq that that $E_Y[\log P_{Y|\hat{Y}}]$ is the average cross entropy of the output, which I do not believe is true: you'd have to use $P_{\hat{Y}|Y}$ and take the expectation wrt \hat{Y} for that to be true.
> - Equation 12 is equating distribution-level quantities to empirical quantities obtained from the dataset; this is confusing.
> - You use $\hat{Y}$ and $\tilde{Y}$ seemingly exchangeably (along with many other minor formatting issues).
> - The notation $I(X;\hat{X};\tilde{Y})$ (eq 19) is unclear and I am not familiar with "mutual information between 3 random variables".
> - Proposition 5 is once again conflating empirical estimates of probabilities with distribution-level quantities.
> - Equation 22 uses $\hat{x}_i$ out of nowhere, and this quantity does not depend deterministically on the conditioning variable $\hat{x}_j$.
> - Equation 23 uses the notation $H(\tilde{Y}|\hat{X}, \hat{Y}|X)$, which has no meaning.

---

### Review · Reviewer_snbf · 2025-10-03

**Summary Of Contributions:**

**Summary**

The paper introduces a new sampling approach for using diffusion models for dataset distillation. The proposed technique is shown to particularly improve the distillation performance in the low images per class (IPC) setting.

The authors primarily build on the assumption that (a) prototype information; and (b) contextual information are two indicative factors for good distillation performance, and the paper goes into deeper detail about how to derive and incorporate variational estimates for these heuristics into the diffusion model sampling process.

Another key takeaway from the paper is that the relative contribution of the two quantities (prototype / contextual information) strikes a delicate tradeoff with the distillation performance at different IPC levels (one is useful for low IPC vs. the other being useful in high IPC settings).

**Strengths**

- Novel guided-sampling approach for using diffusion models for dataset distillation.
- SoTA results when comparing with reasonable baselines.
- The paper is well-written and easy to follow along.
- Easy to tweak the distillation performance under different IPC settings (via tweaking $\beta$).

**Weaknesses**
- Does not provide theoretical guarantees on how tight are the resulting lower bounds for the proposed heuristic quantities ($I(X;Y)$ and $H(X|Y)$).
- Doesn't study the reliance of the proposed sampling approach on the choice of the pretrained diffusion model. What if the guidance for the data distillation task takes the diffusion model in unexplored trajectories that are extremely out of distribution relative to the pretraining task (in principle, it should?)

**Audience:**

Yes

**Audience Explanation:**

Yes, the paper would prove a worthwhile read for the data distillation and diffusion models communities.

**Claims And Evidence:**

Yes

**Claims Explanation:**

While the paper could improve by having more theoretical guarantees and more ablation studies (see the weaknesses outlined above), I still feel the paper has enough evidence to be a meaningful contribution to the data distillation literature.

**Requested Changes:**

Please see the two weaknesses listed in my earlier comment. While more results on both of these things would be nice to have, I wouldn't mark them as critical for recommending acceptance.

---

> ### Author Response · Authors · 2025-10-14
>
> Thank you for your positive feedback. To the best of our knowledge, we are the first to decompose the information in the dataset into prototype information and contextual information in an information-theoretic manner in the literature. As you noted, another key contribution is our analysis of the varying optimal trade-off between the contextual information and prototype information under different IPC settings. Please find our response to your questions below.
>
> **W1. Does not provide theoretical guarantees on how tight are the resulting lower bounds for the proposed heuristic quantities**
>
> Thank you for your helpful comments. We have added Remark 2 to explicitly state the conditions under which our variational estimators become tight. Specifically, when the encoder’s mutual information is maximized and equal to the entropy of the samples H(X), the bound is tight.
> In general, it is challenging to estimate the tightness of the lower bound on real-world datasets, as the original dataset distribution is generally unavailable, which is the motivation for proposing the variational estimator to estimate both information quantities.
> We empirically quantify tightness on a synthetic Gaussian dataset where ground-truth I(X;Y) and H(X|Y) are available.  As reported in Sec. 5.7, our variational estimator yields substantially tighter bounds than a cross-entropy-trained baseline across all settings. Supporting the effectiveness of our approach.
>
> **W2a. Doesn't study the reliance of the proposed sampling approach on the choice of the pretrained diffusion model.**
>
> Thank you for raising this point. We evaluated IGDS with a stronger pretrained diffusion prior, Minimax-DiT (DiT-based). These experiments, originally in Appendix E, now appear in Section 5.6. The results indicate that IGDS is orthogonal to the choice of prior; when combined with a stronger prior distribution, the classification performance improves consistently, outperforming the baselines across all the IPC settings.
>
> **W2b.  What if the guidance for the data distillation task takes the diffusion model in unexplored trajectories that are extremely out of distribution relative to the pretraining task (in principle, it should?)**
>
> Thank you for the thoughtful question. In principle, strong guidance could steer sampling off the base model’s data manifold. Empirically, we observed no failure cases during the sampling: IGDS consistently produces coherent and meaningful distilled images across all runs. This is consistent with the standard assumption in generative dataset distillation that the pretrained diffusion model’s support overlaps the target dataset’s distribution; we now state this explicitly in the Introduction and Methodology.

---

### Decision · Action_Editor_nDcR · 2025-11-23

**Recommendation:** Reject

**Audience:**

Yes

**Audience Explanation:**

Dataset distillation remains an active research area, and the idea of using information-theoretic objectives to guide diffusion-based synthetic data generation is conceptually appealing. The empirical performance across IPC regimes and the discussion of prototype versus contextual information provide potentially useful insights for researchers working on generative distillation, efficient training, or information-theoretic learning. However, despite the topical relevance, the clarity and correctness issues in the core theoretical formulation substantially limit the paper’s impact in its present form.

**Claims And Evidence:**

No

**Claims Explanation:**

This paper introduces an information-guided diffusion sampling (IGDS) framework for dataset distillation in image classification. The core conceptual idea is to decompose dataset information into two components, mutual information $I(X;Y)$ (“prototype information”) and conditional entropy $H(X \mid Y)$ (“contextual information”), and train variational estimators for both, which then guide a diffusion model during synthetic data generation. On the empirical side, the method achieves competitive or state-of-the-art downstream performance across multiple IPC regimes on Tiny ImageNet and ImageNet subsets, and the experimental evaluation is broad and carefully executed.

However, the central derivations underlying the variational objectives contain significant issues that raise concerns about correctness and conceptual soundness. As highlighted in detail by Reviewer GygP, several key steps conflict with basic information-theoretic identities, for example, the inequality in Eq. (6) would imply $I(\hat{Y};Y) \ge H(Y)$, contradicting fundamental bounds. Similarly, Eq. (23) introduces the expression $H(\tilde{Y} \mid \hat{X}, \hat{Y} \mid X)$, which is not valid conditioning notation for random variables, is not defined anywhere in the manuscript, and has no clear probabilistic interpretation. Beyond this, the sequence of identities in Eqs. (19)–(21) relies on a triple mutual information term $I(X;\hat{X};\tilde{Y})$ that is not defined. These are not isolated typographical slips but structural issues in the theoretical development.

Furthermore, the probabilistic setup remains ambiguous: the manuscript frequently shifts between population-level and empirical quantities, uses inconsistent expectation notation, and invokes constructs without a precise definition. Reviewer 7Cq1 also noted shortcomings in the formulation of the random variables and distributions. In the revision, these concerns were addressed only partially, and the problematic equations, including those mentioned above, remain in the current version.

Because the main technical contributions of the paper rest precisely on these variational estimators, the unresolved issues materially limit confidence in the validity of the proposed information-theoretic framework.